# A Review of Multi-Modal Learning from the Text-Guided Visual Processing Viewpoint

**DOI:** 10.3390/s22186816

**Published:** 2022-09-08

**Authors:** Ubaid Ullah, Jeong-Sik Lee, Chang-Hyeon An, Hyeonjin Lee, Su-Yeong Park, Rock-Hyun Baek, Hyun-Chul Choi

**Affiliations:** 1Intelligent Computer Vision Software Laboratory (ICVSLab), Department of Electronic Engineering, Yeungnam University, 280 Daehak-Ro, Gyeongsan 38541, Gyeongbuk, Korea; 2Department of Electrical Engineering, Pohang University of Science and Technology, Pohang 37673, Korea

**Keywords:** Text-to-Image, Text-to-Visual, computer vision, neural networks

## Abstract

For decades, co-relating different data domains to attain the maximum potential of machines has driven research, especially in neural networks. Similarly, text and visual data (images and videos) are two distinct data domains with extensive research in the past. Recently, using natural language to process 2D or 3D images and videos with the immense power of neural nets has witnessed a promising future. Despite the diverse range of remarkable work in this field, notably in the past few years, rapid improvements have also solved future challenges for researchers. Moreover, the connection between these two domains is mainly subjected to GAN, thus limiting the horizons of this field. This review analyzes Text-to-Image (T2I) synthesis as a broader picture, Text-guided Visual-output (T2Vo), with the primary goal being to highlight the gaps by proposing a more comprehensive taxonomy. We broadly categorize text-guided visual output into three main divisions and meaningful subdivisions by critically examining an extensive body of literature from top-tier computer vision venues and closely related fields, such as machine learning and human–computer interaction, aiming at state-of-the-art models with a comparative analysis. This study successively follows previous surveys on T2I, adding value by analogously evaluating the diverse range of existing methods, including different generative models, several types of visual output, critical examination of various approaches, and highlighting the shortcomings, suggesting the future direction of research.

## 1. Introduction

Artificial intelligence, specifically neural networks, is the recreation of human intelligence. The primary goal of neural networks is to sense various surrounding stimuli, understand raw data, and interpret meaningful results in a similar manner to humans. In order to match or surpass human intelligence, machines must be able to analyze and correlate multiple domains of data, such as visual, auditory, and natural language. Therefore, over the past few years, researchers have shifted their focus to the concept of learning cross-domain data interpretation.

More importantly, humans’ innate ability to visualize pictures and provoke imagination from natural language, also known as “seeing with the mind’s eye” [1] is a crucial aspect of cognitive function. A few years ago, it was unbelievable that machines could interpret natural language or even execute intelligent visual tasks. Thus, in the beginning, some researchers tried using traditional hard-coded Al techniques [2], which suffer from several drawbacks such as inconsistency, low quality, lack of diversity, and handcrafted algorithms, but the advent of extraordinary neural networks turned myth into reality, especially the generative models such as GANs and VAE. These models are capable of generating unseen plausible images automatically. Similarly, the field of natural language flourished as researchers understood this phenomenon to pass it to machines using various AI techniques.

Although the idea of multimodal learning stemmed from [3,4], proposing the conditioning of additional variables on generative models, the dynamic generation of images or videos from natural language remains an unsolved problem due to the lack of semantic correlation between language as text and visual domain. Aiming at this gap, Manismov et al. [5] introduced alignDRAW, an extended version of Deep Recurrent Attentive Writer (DRAW) [6], that can generate images from captions. Since recurrent neural networks and autoencoders had limitations, Reed et al. [7] in 2016 made the first attempt at T2I utilizing the power of a generative network, GAN, following which T2I received considerable attention.

T2I has made significant progress in the last 5 years. Thus, several studies [8,9,10,11] have put forth a semantic taxonomy for adversarial text-to-image synthesis (T2I), summarizing the efforts made using GAN [12] mainly. In contrast, this paper focuses on two primary gaps in previous studies. First, we complement the previous work on GAN-based T2I by revising the list of GAN models with current state-of-the-art (SOTA) techniques while providing an in-depth review and a comparative analysis with the previous ones. Second, we conceptualize T2I as a broader research area, Text-guided Visual-output (T2Vo), which is a parent node with three significant subcategories: image, story, and video. Depending on the output task, dimension, and method, each category is further subdivided into generation or manipulation, 2D or 3D, and simple AI or deep learning, where our focus is on deep learning. Additionally, as a continuation of previous T2I reviews, our emphasis is on other T2Vo tasks, i.e., story and video, and generative visual models other than GAN, such as auto-regressive and VAE.

Based on this critical analysis of text-visual cross-modality for generating visual output, we present an outline of the current research direction with existing defects that require further attention by the community. Moreover, we discuss new avenues of research in this domain, ranging from improved datasets to the discovery and refinement of various generative models with more reliable assessment criteria.

Even though cross-domain learning is a wide field of research, this paper attempts to comprehend only the text-guided visual output, as shown in Figure 1. As the scope of applications and categories increases, it becomes increasingly difficult to identify new directions and gaps without a comprehensive record of previous research. Our contribution, therefore, is threefold:Viewing T2I as a vast domain, we comprehensively present a semantic taxonomy of Text-guided Visual-Output (T2Vo) contrary to either text-to-image synthesis [8] or T2I using GAN exclusively [9,10,11].We comparatively analyze previous and new SOTA approaches over conventional evaluation criteria [11] and datasets by paying particular attention to the models missed by earlier studies.By critically examining different methodologies and their problem-solving approaches, we can set the stage for future research that can assist researchers in better understanding present limitations.

In the real world, data exist in various forms, also known as modalities. These are often found in textual and visual content with a lengthy application history in AI. Therefore, studies related to multimodal learning [13,14] also resemble our work. However, due to the broad spectrum, surveys on multimodal learning, explicitly targeted at the text-to-visual output, are scarce and lack in-depth analysis. To the best of our knowledge, there is no comprehensive research on text-to-visual output.

For a thorough understanding of text-guided visual output, an understanding of the fundamentals is necessary. So, Section 2 lays the foundation for T2Vo, followed by the selection criteria mentioned in Section 3 to narrow down the domain. We then provide the broad-spectrum semantic taxonomy for T2Vo detailed in Section 4, Section 5 and Section 6. Next, Section 7 summarizes the different datasets mentioned in the literature, using which Section 8 enlists the evaluation metrics commonly used in these studies. Based on these profound approaches to intelligent tasks, Section 9 lists the current industrial and future applications for T2Vo. After applications, we highlight the current challenges in Section 10. Finally, Section 11 describes the future directions for T2Vo through an analysis of current SOTA methods, then we discuss some key insights and conclude the paper in Section 12.

## 2. Foundation

This section discusses the critical components to understanding the concept of T2Vo, mainly the image and text models, which are then combined to form the joint representation.

### 2.1. Language and Vision AI

#### 2.1.1. Language Models

Natural language is amongst the most common methods of communication, so training machines to understand and communicate in natural language with humans is essential for machine–human interaction [15]. Before the deep learning era, considerable efforts were made for natural language processing (NLP), manipulating only traditional AI techniques such as rule-based approach [16] or simple machine learning techniques [17]. A primary drawback of utilizing the carefully designed hand-crafted features for application-specific algorithms is that there is limited intelligence, which cannot handle large amounts of data. Therefore, this has resulted in the need for neural networks, which feed on data and computational power, to deal with such complex data [18].

For assimilating T2Vo, frequently used NLP models, RNN, LSTM, GRU, Transformer, GPT, and BERT need to be revisited before Section 3. However, conceptualizing such models first requires an understanding of NLP’s core concepts, spanning three primary divisions: feature representation, seq2seq framework, and reinforcement learning [18].

Text analysis can incorporate various forms and levels of features that represent meaningful and desirable information. The process of extracting information from a corpus consisting of several paragraphs with sentences created from a semantic combination of words can be complicated, making it imperative to learn distributed representations. Therefore, depending on the application under consideration, text features exist at different levels [19], from characters [20] or symbols to words [21], sentences [22], paragraphs, and documents, as shown in Figure 2. Furthermore, text conversion to a form readily acceptable by machines, text vectorization, otherwise known as text embeddings, extends from simple, such as one-hot [23], BoW [24], CBOW [25], WCBOW [26], and TF-IDF [27], to more complex models utilizing neural nets, such as word-level [25,28,29], subword-level [30], and character-level [23] encoding. These representations typically serve the purpose of capturing the semantic and syntactic context or word to differentiate between other corpora.

Another key concept for NLP is the Seq2Seq framework, which usually involves an encoder–decoder design by implementing RNN [35], LSTM [36], GRU [37], or CNN [38] cells, more recently replaced by Transformer [39]. The input and output of this framework are considered as a sequence. The workflow of RNN and its variants such as GRU and LSTM are the same, i.e., processing information temporally and thus memorizing previous states for future predictions. However, RNN suffers from memory loss and gradient vanishing, so LSTM and GRU prevent this situation by inducing additional gates. LSTM uses a forget-output-memory gate while GRU employs an update-reset gate [40]. Although the sequential processing of RNN models can be leveraged by CNN [41,42], which follows a hierarchical architecture for parallel processing, it fails to effectively capture the dependencies among different combinations of words [43]. So, in short, LSTM and GRU mitigate distant information loss using recurrence, but their sequential nature inhibits them from parallel computation, whereas CNN is impractical for long-term dependencies. Consequently, these constraints led to the development of the most advanced model, called Transformer—an approach to sequential processing by eliminating recurrent connections and introducing a self-attention module [44]. This model garnered much attention as it defined a new state-of-the-art, laying the foundation for future deep learning architectures such as BERT [45] (by Google) and the GPT series [46,47,48] (by OpenAI). BERT (Bidirectional Encoder Representations from Transformers) and GPT (Generative Pre-trained Transformer) are pre-trained models trained on massive unsupervised data for multiple downstream tasks with discriminative fine-tuning on application-specific data. GPT followed the Transformer decoder and only applied unidirectional training. Comparatively, BERT’s distinguishable features are focused on the Transformer encoder architecture with Bidirectional training.

Finally, Reinforcement Learning (RL) is another approach in which an agent learns how to choose an action correctly within a particular environment to maximize rewards. It is effectively applied in NLP to mitigate two primary issues faced in the seq2seq framework: exposure bias and training–testing inconsistency [18]. Exposure bias occurs due to the optimization objective via Teacher Forcing [49], which, during training, uses the previous state and ground truth as inputs for the decoder to generate the current state. However, at testing, it relies only on the previous state and induces an error growth, handled in [49] with scheduled sampling as a solution. Moreover, the training–testing inconsistency is forced when non-differentiable measures such as METEOR [50] and ROUGE [51] are used for evaluation. So, the use of RL in NLP has recently shown excellent potential [52], among which actor–critic models [53,54] and policy gradient techniques [55,56] are commonly used.

Initially, SEARN [57] used model predictions for seq2seq modeling and generation during training, which is categorized as reinforcement learning and trains the policy to predict optimal action from a sequence of actions. In addition, actor–critic training works slightly differently: the actor is a network that generates output while the critic model estimates its performance. However, a critical problem in using RL for NLP is the immense action space responsible for slow training and difficulty in the correct action selection.

#### 2.1.2. Visual Models

In addition to text models, visual models and their basics hold equal significance for a thorough knowledge of T2Vo. The study of digital images and their processing is a well-established topic with tremendous research in the past. Similar to text, semantic representation of raw pixel values, also called features, is necessary to derive meaningful results for machine processing. This task of feature representation can be as simple as splitting white and black colors based on some threshold and as complex as representing complicated objects in the medical field, such as radiology [58]. Based on the level of processing, vision can be low-level, such as segmentation and edge detection, or high-level involving machine learning, including Image classification [59], object detection [60], and Image generation [61]. High-level vision in machine learning, especially Deep Learning, can be either discriminative or generative, as represented in Figure 3.

**Discriminative models** usually learn image features via training from a labeled dataset, referred to as supervised learning, and often serve as feature encoders. Among these models, convolutional neural networks (CNNs) have provided a significant contribution in presenting an operative class of models to understand the content present in an image better, resolving different realm problems in recent decades. Their effectiveness on images is due to the convolutional layers followed by pooling layers, which reduce the number of parameters to converge faster. Thus, the CNN reduces dimensionality by exploiting context information in a small neighborhood of pixels [62]. In this way, cascaded convolutional layers can learn complex features by stacking simple learned features at multiple stages.

The first CNN model is LeNet-5 [63], trained to recognize handwritten characters and utilizes convolutional, pooling, and fully connected layers. This pioneering work is moderately immune to shift, scale, and distortion. However, it still struggles to surpass the traditional SVM algorithms.

Following LeNet-5, in 2012, CNN called AlexNet [64] expanded the basic idea by proposing a deep and wide network having more convolutional layers with ReLU as an activation function to mitigate the gradient vanishing problem. It harnessed the power of GPUs for the first time. Additionally, they use dropout and data augmentation to avoid over-fitting along with max-pooling to reduce blurriness. Furthermore they employed LRN for normalization.

Although AlexNet provided a solid foundation to apply CNN models for the images but failed to explain the relationship between the depth and performance of the model. Therefore, VGGNets [65] prove that performance is moderately related to the depth of the model. So, the results were improved by increasing the depth of the model without LRN and a small kernel size (3 × 3) while having reduced model parameters.

The earliest work on a large-scale CNN model was by GoogleNet [66]. These extensive models came into existence by stacking multiple Inception modules for the first time, which they split into four versions. In Inception-v1, multi-size convolutional kernels were introduced to reduce the computational cost while extracting multi-scale feature maps. Afterward, Batch normalization was employed in Inception-v2 [67] to solve internal covariate shift problems while increasing the robustness. Then to increase depth and non-linearity in Inception-v3 [68], the RMSprop optimizer is introduced, and the factorization of (5 × 5) kernel to (1 × 7, 7 × 1) and (3 × 3) to (1 × 3, 3 × 1). Inception-v4 [69], however, is based on the ResNet structure, which further extends the depth of these networks and improves their performance; it is also known as Inception-ResNet.

Previous studies have proved the superior performance of deep networks compared to shallow ones. However, they cannot exceed a specific limit due to gradient vanishing and exploding. The introduction of ResNet [70], in 2016, proposed a 34-layer neural network with a crucial contribution of residual blocks, consisting of 2 or 3 layers with bypass connections. After this, many studies followed ResNet and achieved better results, including wide ResNet [71] and ResNet in ResNet [72].

Although Xception [73] came just after ResNet-50, inspired by Inception’s architecture, they replaced the standard Inception modules with depthwise separable convolutions. It was accomplished by simply performing depthwise convolution with a filter of (3 × 3) or (5 × 5) to all channels, followed by pointwise convolution across channels with a (1 × 1) kernel. Moreover, this model also made use of residual connections proposed in ResNet. This slight modification enhances the efficiency with the same parameters as Inception-v3.

To summarize the relation between Inception and Resnet, the first reduces the computational cost by going broader, while the second focuses on computational accuracy by going deeper. Different from these two, another method of efficient feature learning is by resolution scaling at the expense of high computational cost, which unfortunately is not stable for large networks as it immediately lowers accuracy gains. Conclusively, the latest work on CNN is the EfficientNet [74] exploring compound scaling by a compound coefficient to obtain the best from all dimensions: depth, width, and resolution. This baseline network results from Neural Architecture Search (NAS), followed by a family of models, EfficientNets, achieving the best results from models that are about 6× faster and 8× smaller.

To augment EfficeintNet, Noisy Student training [75] proved its importance. This idea is the fusion of self-training and distillation with added noise for student training. Compared to the Knowledge Distillation [76], the innovation of this model is knowledge expansion. In this way, it surpasses the teacher with the help of a more challenging environment, noise, by an iterative process that learns a larger student model from the trained teacher model.

Capsule networks [77] are implemented to solve CNN problems, spanning localization, information loss, and low 3D viewpoint variation. These problems arise because of the dependency on local pixel groups and the pooling layers. In the most high-level notion, instead of forwarding individual neuron activation from one layer to another, as in CNN, capsule networks represent each capsule as a small nested neural network that outputs a whole vector. This full-length vector encodes the probability of detecting a specific feature, where the direction helps define the state of the feature, e.g., location, pose, and scale.

Recently, the exceptional progress and attention to Transformers in NLP changed the focus of the computer vision researchers to adopt new models to vision tasks, such as DETER [78]. However, until last year, no work proposed the direct use of Transformers to vision tasks except Vision Transformer (ViT) [79]. The reported performance of this model is remarkable, especially with an 80% decrease in training time with comparable accuracy from the best CNN models. A reasonable explanation of this improvement is the direct use of image patches rather than filtered data of a small area of pixels from an image, ignoring the relationship between parts of the image, thus losing some valuable information.

A detailed summary of CNN models and their evolution in the last few years have been presented by other studies [62,80]. Based on these reviews and the combination with the latest techniques, we present a brief hierarchy of the deep discriminative models in Figure 4. The latest techniques such as Caps-Net [77], Noisy Student training [75], and Vision Transformer (ViT) [79] are in the maturation phase and are less studied for T2Vo tasks.

**Generative models**, unlike their counterparts, assume the signal to be deterministic, which is obtained from a defined transformation on some latent variable [81] to learn unsupervised data. Generally, generative models are classified as cost function-based and energy-based models. Generative adversarial networks (GANs) and Autoencoders are cost-based generative models, while Boltzman Machine, its variants, and Deep Belief Networks (DBFs) are energy-based models. However, according to the study by Ian Goodfellow [82], the generative models derived from maximum likelihood are distinguished based on their representation. To combine both ideas, Figure 5 shows the classification of generative models. Since most of the work in the underlined topic is based on GAN with minor achievements in other generative models. It implies the elaboration of GAN and other principal generative models to highlight their key features, exploiting the pros and cons.

Among the most famous generative models, the earliest work utilized the Boltzmann machine (BM) [84] in 1983 to find the best combinations of hypotheses satisfying the input data constraints. After this, there are series of implementations advancing the idea, namely Binary Boltzmann machine [85], Restricted Boltzmann Machine (RBM) [86], Deep Belief Networks (DBN) [87], and Deep Boltzmann Machine (DBM) [88]. Theoretically, all these models can learn complex distributions, but practically BM suffers from tractability problems. So, RBM was designed to resolve it, and its advanced version is called DBM, which has multiple layers trained in two stages: pre-training and fine-tuning. Figure 6 shows the structure of BM, RBM, DBN, and DBM in comparison.

One of the most challenging limitations of the Boltzmann machine is its extension, for which the variational autoencoder (VAE) [89] implemented a directed model purely trainable with gradient-based methods. VAE is a modification of Autoencoders [90], utilizing encoder–decoder architecture for recreating input at its output while reducing the dimension. However, it should not learn identity function but instead learn underlying patterns of data distribution for generating new data. So, the sole intention of VAE is to train a parametric encoder producing distribution parameters by learning code layer distribution, assuming it follows Gaussian distribution. As a result, we obtain noisy data unsuitable for many applications.

The most explored generative model since 2014 is GAN [12]. This model derives from the cost function following a minimax 2-player game modeled as zero-sum, where the absolute difference of rewards is minimal, to help learn both simultaneously. At its core, there are two networks, Generator *G* and Discriminator *D*, trying to defeat each other, where G generates data from stochastic noise and D is supposed to distinguish the generated (fake) data from the original one. To sum the idea of GAN, consider G as a differential function, accepting random noise as its parameter to produce data that probably follows the given data distribution, where D is a classifier function to map the data distribution to a probability which defines the likelihood of data to be the actual data. However, training these models is no simple task and requires delicate handling. Generally, learning in GAN is split into two separate but consecutive stages: first, D is trained while suspending G for some epochs; then, D is held so that G can learn by mistakes, and vice versa. Although the best results are from these models, they are held back due to the restrictions of hard training, divergence trap, insignificant occurrences, 3D perspective, Global structure of images, and finally, Mode collapse, which is the worst of all.

### 2.2. Joint Representation

T2Vo requires the semantic concatenation of language and visual data, which is not a trivial task. Therefore, a common multimodal representation of the two domains is necessary. In terms of multimodal representations, two types of divisions exist [14], joint and coordinated. For learning joint distribution, unimodal signals are defined into the same space, whereas the signals processed separately followed by enforced similarity constraints converge onto the coordinate space. The first division, joint representation, is generally suitable for applications where multimodal data resides for both training and testing, which is mostly the case of T2Vo. Additionally, most of the work on T2Vo represents the two data distributions in the same space. So, we keep our attention only on this type.

Early approaches to joint representations acquire conventional methods briefly discussed in [91]. However, the deep learning methods for this task either use graphical models, neural networks, sequential models, or generative models, as shown in Figure 7. We adopt this classification from [91] and modify it to highlight text-visual multimodal representation exclusively. In T2Vo, visual data is generally dealt with CNN models, whereas sequential models are responsible for the text. In Figure 7, the highlighted parts indicate predominantly used models for text-to-visual multimodal learning.

## 3. Text-Guided Visual-Output

To be concise, the vast domain of text-to-visual output requires a careful selection of papers and a thorough study to merge and relate various proposed ideas, distinguishing the main contributions from minor improvements. Therefore, initially, we introduce the paper selection procedure, inclusive of selection criteria, methods of selection, and screening procedures.

To begin with, we narrowed the domain of our search by only focusing on text-to-visual output, more precisely, augmenting the visual field with the help of text, not the other way around. Additionally, a targeted search from top-tier journals and conferences is a vital source to view the literature. To further decrease the span, the center of attention is the progress based on deep learning techniques, roughly originating from 2009. Nonetheless, considering the whole notion, we have also briefly introduced the earliest work.

Once the tentative rules are defined, the next step to our paper selection procedure involves search strategies, for which we adopted two methods: search engines and the related work section of SOTA methods. Explicitly, research article databases such as IEEE Xplore, ACM Digital Library, arXiV, and Papers With Code (PWC) helped identify the trending research in the specified domain, leading to previous SOTA methods. Besides these databases, some manually selected top-tier conferences related to Artificial Intelligence and Machine Learning also played a vital role in the selection process, mentioned in Table 1 specifying the elected number of publications as well. Finally, the papers deemed potentially relevant are put through screening to evaluate them as being inclusive or exclusive of thorough analysis, indicating significant changes.

In this study, we present a broad-spectrum taxonomy for text-guided visual results. First, we categorize it into image, story, and video based on the consistency of the generated output. Next, we distinguish these categories in terms of complexity from a dimensionality viewpoint, which is further subject to either generation or manipulation depending on the input. After the broad division of visual output, we further cluster different models based on the approach used for producing the output. Since the study focuses on deep learning techniques for T2Vo, we mainly concentrate on the generative models rather than the retrieval or conventional ones. These deep generative models belong to one of the four classes, GAN, VAE, Auto-regressive, and energy-based models. However, for the sake of wholeness, we shortly mention the retrieval methods as well. For the semantic clustering within each class, we critically evaluate the models and point the significant contributions along the way. Additionally, we specify the efforts made for minor improvements to the relevant area. The proposed taxonomy of T2Vo is shown in Figure 8. Table 2 shows the summarized characteristics of the selected studies in Figure 8.

## 4. Image (Static)

In the underlined subject of T2Vo, the most studied topic in the recent few years is text-to-image (T2I) generation; therefore, an extensive amount of research is devoted to this task. Consequently, we can now generate more appealing and realistic images from text. In our proposed taxonomy, we divide this task into 5 different categories depending on the type of model used for generation.

### 4.1. Energy-Based Models

Models under this category mainly rely on generating images from conditioned energy-based generative models, chiefly on variants of Boltzmann machines.

The initial work of Xing et al. [92] to model a joint distribution between images and text from an energy-based generative model is through the use of the multi-wing harmonium model, utilizing a two-layer random field, which is considered as a form of Restricted Boltzmann Machine (RBM). This RBM model uses Gaussian hidden units combined with Gaussian and Poisson visible units, and learning is performed by a derived contrastive divergence and variational algorithm. However, this model is too shallow to learn various data modalities with different statistical properties. So, the model only generates results for classification and retrieval.

Nitish et al. [93], intending to deal with distinct statistical properties of multi-modal data, uses a separate 2-layer Deep Boltzmann Machine (DBM) for each modality as a generative model for obtaining a joint representation by combining features across modalities. For image–text bimodal DBM, the Gaussian model represents image features and a Replicated Softmax model for text features over word count. In this way, sampling from conditional distributions allows the model to learn representations even when some data modalities are missing. The experimental results on image–text and audio–video data represent the capability of this model as a classification or retrieval and still struggle for generation tasks.

Previous models on Conditional Random Fields (CRF) for text–image modality are limited to labels as text, whereas the text in natural form comprises sentences containing information about objects, attributes, and spatial relations. For this reason, Lawrence et al. [94] explore learning visual features corresponding to semantic phrases derived from sentences. From sentences, extracted predicate tuples of two nouns and one relation along with CRF [262] formulation with nodes as objects and edges as the relation form a scene. Since the goal is to relate images with sentence-based text, scene generation is still retrieval-based with the invention of a new dataset named abstract scenes.

Generally, multi-modal representation learning involves learning joint representations on top of model-specific network layers, as in [92,93]. However, it cannot reason about missing data in the presence of the rest, highlighting the insufficient association between different modalities. Therefore, improving joint representation learning for multi-modalities through deep generative models is the goal of Kihyuk et al. [95]. They suggested a novel multi-modal representation learning framework trained to maximize the variation of information rather than maximum likelihood. The use of the Multi-modal Restricted Boltzmann Machine (MRBM) with new contrastive divergence and multi-prediction training algorithms helped test this theoretical insight. Furthermore, the model extended with a deep recurrent network for finetuning achieved a significant performance on the visual recognition and retrieval task.

### 4.2. Auto-Regressive Models

Autoregressive (AR) models are feed-forward sequential models and predict future values based on the past ones. Unlike RNN, the past values act as input rather than the hidden state. Due to this, it applies to data having some correlation between values in time series and among one another. We group the models which employ this approach without any other generative model such as GAN or VAE under the autoregressive models for text-conditioned image generation.

#### 4.2.1. Generation

**CNN-based:** Van et al. in [263], proposed PixelCNN and PixelRNN as generative models for modeling image distribution through a deep neural network that sequentially predicts pixels in an image. Built on this theory, the continual work of Van et al. [96] introduced conditional image generation based on PixelCNN architecture as a pioneering image density model. This new model combines the individual strengths of speed from PixelCNN and performance from PixelRNN to a gated variant of PixelCNN, Gated PixelCNN. The conditioning vector for Gated PixelCNN can either be labels, tags, or latent embeddings from other networks. Furthermore, this model shows excellent capability as an image decoder in an autoencoder.

Following the approach of Van et al. [96], Reed et al. [97] implemented caption-conditioned image generation from Gated PixelCNN to compare its performance with the GAN-based generative model [43]. Apart from text conditioning, additional condition on part key-points and segmentation masks resulted in the controlled generation of images. In this improved model, a character-level text encoder and image generation network are jointly trained end-to-end via maximum likelihood.

In PixelCNN, although training is fast, costly inference requiring one network evaluation per pixel limits its use for practical implementation. A joint work by Reed and Van [98] highlighted this constraint and proposed parallelized PixelCNN for more efficient inference. In this variant, by modeling a specific group of pixels as conditionally independent, the new PixelCNN model achieved competitive density estimation and was orders of magnitude faster. The main design principle follows a coarse-to-fine ordering of pixels. Due to the new conditional independence structure, generating higher-resolution images up to 512 × 512 is possible. As tested before, the conditioning is either on class, caption, or layout with an additional task of action-conditioned video generation.

Pursuing human–machine interaction by grounding language into perception and action, Xinlei et al. [99] created the CoDraw dataset based on abstract scenes. They formed this dataset through a collaboration between a human teller and a drawer, aimed to generate semantically rich scenes from the dialog-based language in an interactive way. Initially, two human players played the game of telling and drawing, but for automation, agents based on one of the two methods, rule-based or neural-based, performed the task on the collected dataset. So, utilizing a bidirectional LSTM, the neural drawer encodes text and then uses a feed-forward network to create a scene. On the other hand, the teller uses Reinforcement learning on LSTM for generating captions.

Similar to CoDraw, Text2Scene [100] also generates scenes from natural language, but unlike CoDraw, which uses chat logs, the language is sequential captions for progressively generating an image. The model consists of a text and image encoder for obtaining sequential input representation and current image state, respectively. Next, a convolutional-recurrent module keeps track of the already generated scene, followed by two attention-based predictors that sequentially focus on different parts of the text to decide about object type and location. Optionally, a foreground embedding step determines the appearance for patch retrieval in synthetic image generation. The authors showed the model, under minor modifacations, can generate different forms of scenes, including cartoon-like, natural, and synthetic ones.

**Transformer-based:** Another study, focused on zero-shot learning for T2I, trained a 12-billion parameter autoregressive transformer on 250 million image–text pairs [101]. The authors named it DALL-E, and it follows a two-stage training procedure due to the computational limits. In the first stage, a discrete VAE is trained to compress images into a grid of image tokens, whereas stage two concatenates the image–text tokens and learns an autoregressive transformer to model the joint distribution of the text–image pair. We can visualize the overall procedure as maximizing evidence lower bound (ELB) [89] on the joint likelihood of the model distribution over images, captions, and tokens.

Similar to DALL-E, CogView [103] is another pre-trained model for text-image pairs. However, this transformer-based model has 4 billion parameters after training on 30 million high-quality Chinese text–image pairs, where the images are compressed by a trained VQ-VAE [264]. Compared to DALL-E, pretrained CogView is finetuned to apply on downstream tasks, such as image captioning and super-resolution. Additionally, this model enables self-reranking for post-selection to avoid the CLIP [224] model as in DALL-E, with a new evaluation metric, called caption loss, to measure quality and accuracy for text–image generation. For stabilized training of a large-scale transformer, two techniques, PB-relaxation and Sandwich-LN, are also utilized to eliminate overflow in forwarding.

Earlier autoregressive models incorporate the image in a linear 1D order, which is unidirectional and overlooks large parts of the scene until generation, and process the entire image on a single scale, thus ignoring more global contextual information. From these observations, recently, a more advanced version of the autoregressive model for a variety of tasks, including text-to-image synthesis and image inpainting, was presented in ImageBART [105]. As a remedy for the mentioned problems, this model incorporated a coarse-to-fine hierarchy of context by combining autoregressive formulation with a multinomial diffusion process. Specifically, first, a multistage diffusion process [265] coarsens an image by successively removing information to learn a compressed image representation, which then is inverted by a trained short-Markov chain. Individual transition probabilities from this chain form an independent autoregressive encoder–decoder model based on transformer architecture [39].

**Distillation networks:** Although the models discussed in this section are auto-regressive, being sequential, some outliers lacking any other generative model for text-to-image synthesis also fall under this category due to the use of the deep CNN model for image generation.

The first study to utilize knowledge distillation for text-to-image generation uses a symmetrical distillation network (SDN) [106]. This model visualizes T2I issues in two gaps, heterogeneous and homogeneous. To exploit this, a generic discriminative model, VGG19, guides the training of a generative model on a high level for bridging the text–visual heterogeneous gap and a mid-to-low level for realistic images as the homogeneous gap. The target generative model, the student, is symmetrical to the source discriminative model, the teacher, with two-stage training exploiting coarse-to-fine learning.

The authors of SDN further extended their method in [107] for text-to-image synthesis (T2IS). In this extension, two main contributions include knowledge distillation from two models, classification and captioning, for T2IS, and a multi-stage distillation paradigm for adaptation to various source models. Practically, they added a third distillation from the captioning model, following [108] with Inception-v3, over the first two from the classification model, VGG19.

#### 4.2.2. Manipulation

The manipulation task deals with the type models, which can understand the provided input, where, based on some given condition, they can modify the required part. Autoregressive models also share this capacity to manipulate a given image from user-provided text.

Language-based image editing (LBIE) [204] initiated the use of a neural network for image manipulation, specifically, a generic framework for modeling two subtasks, segmentation and colorization. The framework uses recurrent attentive models with a termination gate for each image region to dynamically decide to continue extrapolating additional text information after every step. At a high level, the model comprises a deep CNN as an image encoder and a bi-LSTM with GRU cells as a text encoder, on top of which there is another LSTM with attention for fusion between text-image features through termination gates. For evaluation, a newly created dataset, named CoSaL, is used for experimentation with two other datasets. On the oxford-102 flower dataset, this study is the pioneering work to perform colorization.

Image editing, until language-driven image editing (LDIE) [205] explored either image retouching operation without text input [266,267] or text-guided manipulation of simple object-centered images [169,172,210,268]. More importantly, language-based single editing tasks, such as retouching [226] or recoloring [204] also exist. Additionally, a model for text-based image editing, PixelTone [269], is also present in the literature. However, it requires detailed voice instruction with manually selected image regions. Therefore, LDIE is the first study to incorporate language-driven image editing at both local and global levels, where every editing operation acts as a sub-module that can automatically predict operation parameters. To solve the LDIE task, the authors created a new language-driven image editing dataset with editing operation and mask annotations, called Grounded Image Editing Request (GIER). A baseline method applicable to this dataset takes an input image with requests to a multi-label classifier for operation prediction. Next, the operation grounding model outputs the grounding mask for each operation from image, request, and operations. Finally, a cascaded operation modular network generates the final result.

Continuation of the LDIE work, Learning by planning [207], targeted the limitations of GAN-based models for image manipulation, presented by the same authors. They developed a text-to-operation model (T2ONet) for converting text requests to a series of editing operations, guided by pseudo ground truth of possible editing sequences from the target image through a novel operation planning algorithm. Different from their earlier work [205], which needed operation annotation for training, they created an operation planning algorithm to obtain an operation–parameter sequence by comparing input and target images. In addition, they collected another dataset, which they named MA5k-Req, and revealed the connection between pixel supervision and reinforcement learning.

### 4.3. Variational Auto-Encoder (VAE)

Among the most popular generative models, variational autoencoder (VAE) is one. These models learn the posterior distribution P(Y|X) via the Bayesian rule. Explicitly, unlike GAN, VAE learns the likelihood distribution P(X|Y) through loss function. From an architectural viewpoint, the encoder in VAE reduces the dimenstionaly of input data to obtain a latent space with distributions, and through a regularization term, KL-back divergence, on this space, a sample is then obtained from this space to prouce the output through a learned decoder. In this way, VAE maximize the variational lower bound of the loglikelihood. We combine the models that perform image generation by following this technique.

Success on conditional image generation motivated AlignDRAW [5] to generate images from natural language instead of labels through the use of recurrent variational autoencoder with an alignment model over words. This model is the first to initiate text-to-image generation from VAE and is an extension to the DRAW [6] network. Overall a bi-LSTM encodes text and is combined with a latent sequence sampled from prior through inference RNN, given to the generative RNN for creating the final image, which is refined by post-processing using Laplacian Pyramid GAN [270]. The model follows a sequence-to-sequence framework, where the captions and images are sequences of words and patches on canvas, respectively.

Attribute2Image [109] is another study that makes use of VAE for conditional image generation. However, the conditioning is on visual attributes instead of natural language, expressed in terms of multi-dimensional vectors. This work focused on the conditional VAE (CVAE) and proposed a layered foreground–background generative model. The model obtains the posterior inference through a general optimization-based method, applied in the context of image reconstruction and completion.

CVAE-cGAN [110] explored the complementarity of two different generative models, VAE and GAN, for generating high-quality images considered as the composition of foreground and background. This stacking of VAE and GAN facilitates an effective and stable image generation. First, a context-aware conditional VAE (CVAE) designs a text-based basic image layout, with individual attention to the background and foreground. Next, a conditional GAN (cGAN) refines the generated output of CVAE.

To explore, for the first time, the generalization of the VAE framework for T2I, including zero-shot learning, Probabilistic Neural Programming Network (PNP-Net) [111] proposed a modular programmable framework with probabilistic modeling. This approach constructs priors for the generative modeling of complex scenes. The model consists of two core components, first is a set of mapping functions that converts distributions from input over the latent space, such as combine, describe, transform, and layout. Second, a canonical VAE probabilistic modeling framework for inference and learning using the latent space.

The existing autoregressive methods for text-to-image generation suffer from unidirectional bias and accumulated prediction errors, whereas GAN-based methods are limited to simple scenes, for which Shuyang et al. devised VQ-Diffusion [113]. Therefore, based on the vector quantized variational autoencoder (VQ-VAE) [264], where its latent space is modeled by a conditional Denoising Diffusion Probabilistic Model (DDPM) [115], VQ-Diffusion can generate complex images independent of image resolution for efficient computation. The core design of this technique is to model the latent space of VQ-VAE in a non-autoregressive manner, where the mask-and-replace diffusion strategy removes the accumulation of errors.

### 4.4. Generative Adverserial Networks (GAN)

Owing to the property of generating sharp and high-quality images compared to VAE and directly without sequential processing, as in autoregressive models, GAN-based models for T2I are the most studied topic in this domain. Therefore, many studies are devoted to summarizing the advances in GAN-based models for T2I while providing limitations and future directions. Recently, Stanislav et al. [11] proposed an in-depth analysis of GAN-based models for T2I while organizing different works in a reasonable and comprehensible manner. We complement this taxonomy in the following ways:First, we expand over the previous list by adding additional papers and categorizing them into the already given taxonomy.Second, we separate these models into generation or manipulation based on model input.Third, we not only consider the image as 2D but include studies beyond the 2D image, such as 3D images, stories, and videos.

**Generation:** T2I generation is the process of generating images from text. These models take natural language as input and produce pixel space output. However, modeling a joint distribution of text and image for T2I is not a trivial task and hence requires the careful design of a generative model conditioned on text embeddings. For this reason, over the past few years, after the advent of GAN, various GAN-based techniques have been explored, either generating images directly from the text while exploiting the GAN model for improving this task or introducing intermediate supervision for generating better results on complex data. So, we split the T2I generation task from GAN models into two divisions, direct T2I and supervised T2I, which we discuss in the following section.

#### 4.4.1. Direct T2I

Direct T2I methods include the models which directly perform image generation from the text, exploring the capabilities of the GAN model. First, conditional GANs are enlisted as modified GANs to express the introductory T2I task. Second, to improve upon the image quality, stacked architectures are discussed. Quality without text consistency is useless, so we describe attention mechanisms next. Further improvements for T2I utilizing different architectures, such as Siamese, knowledge distillation, cycle consistency, and Memory networks, are then mentioned. Finally, we examine approaches that implement unconditional models for T2I.

**Conditional GAN:** Initially, Mirza et al. [4] proposed conditional GAN for label-conditioned image generation. However, training GAN to find a Nash equilibrium between a generator and discriminator is difficult, upon which Salimans et al. [116] improved the GAN framework through new training procedures and architectural features of feature matching, minibatch, virtual batch normalization, historical averaging, and one-sided label smoothing. An extension to conditional GAN, Reed et al. presented GAN-INT-CLS [7] by conditioning the generator on whole sentence embedding from a pretrained text-encoder. A matching-aware discriminator is trained in GAN-INT-CLS to distinguish between real and synthetic text-image pairs, with three pair types: real-image–matching-text, generated-image–related-text, and real-image–mismatching-text. In addition to the matching-aware discriminator, inspired from AC-GAN [117], TAC-GAN [118] employed an auxiliary classification loss from one-hot encoded class labels. Perceptual-GAN [119], is another advancement over GAN-INT-CLS by introducing perceptual loss in training along with contextual loss and mean-squared error with Frobenius norm.

More recently, a single pair GAN proposed for T2I is shown in DF-GAN [124]. Opposed to other models that utilize a stacked backbone for T2I, mentioned in the later section, DF-GAN can generate compelling images with a single-stage GAN. It does so from a novel deep text–image fusion block in the generator and a target-aware discriminator composed of a matching-aware gradient penalty (MA-GP) and one-way output. Furthermore, the generator is provided with a stable text latent space through a novel approach of skip-z with truncation.

**Stacked GAN:** Simple T2I models [7,118,119] were limited to generating low-resolution images from 64 × 64 to 128 × 128. Therefore, inspired by [270], stacked architectures were applied for T2I. StackGAN [32] is the first to employ a stacked design for T2I, where the first stage generates a coarse 64 × 64 image from noise and text embeddings, and the second stage generates the final 256 × 256 image from the initial picture with encoded text. However, in StackGAN, the model is trained in two steps, for which StackGAN++ [33] improved the architecture via an end-end framework with three generator-discriminator pairs jointly trained for multi-scale conditional and unconditional image distributions with an additional color-consistency regularization term. In addition to coarse-to-fine image generation for high-quality images, both in StackGAN and StackGAN++, conditioning augmentation (CA) is proposed for a smooth conditioning manifold by sampling text embeddings from a Gaussian distribution. Based on this joint training of multi-level generators, FusedGAN [125] utilizes a single-stage pipeline with two generators for unconditional and conditional generation, partially sharing a mutual latent space for training on extensive unsupervised data.

All previous Stacked models for T2I either use a multi-stage GAN framework or multiple generators. So, HDGAN [126] proposed a single-stream generator with hierarchically nested discriminators at multi-scale intermediate layers trained end-to-end to generate 512 × 512 images. This approach is unique in terms of adversarial learning along the generator depth with specific discriminators at different resolutions, trained to distinguish real and synthetic image patches alongside the matching aware pair loss. Hence, the objective function helps generate more consistent images between multiple scales. Similar to HDGAN, PPAN [127] also uses one generator, having a pyramid framework with three distinct discriminators to join strong low-resolution semantic features with weak high-resolution ones through a laterally connected down-to-top pathway. Furthermore, image diversity, semantic consistency, and class invariance are achieved with the help of a pre-trained VGG network-based perceptual loss, image patch loss, and auxiliary classification loss, respectively. In comparison with HDGAN and PPAN, HfGAN [128] employs a hierarchically fused architecture but with only one discriminator. The generation again follows a coarse-to-fine process, where the extracted multi-scale global features from different stages are adaptively fused, requiring only one discriminator. For fusion, following ResNet [71], identity addition, weighted addition, and shortcut connections are adopted.

**Attention Mechanisms:** Focusing on specific input regions is crucial as some components signify more importance than others. Consequently, the attention mechanism by weighing essential segments more allows the network to focus on specific aspects of an input.

Introductorily, AttnGAN [34] incorporates an attention mechanism into a multi-stage refinement pipeline, built upon StackGAN++. This mechanism enables word-based fine-grained details on top of the global sentence vector for T2I through a Deep Attentional Multimodal Similarity Model (DAMSM) loss, where attention is given to the most relevant words for image sub-regions. The DAMSM loss computes the similarity between input text at sentence-level and word-level information with the generated image.

The work of Huang et al. [129] improved the DAMSM loss by introducing true-grid regions inside every bounding box with word phrases, where attention weights depend on the bounding box and phrase information. So, this mechanism extends the regular grid-based attention that utilizes additional phrase features through parts-of-speech tagging besides sentence and word features.

AttnGAN gives attention to each sentence word, which is inefficient. Consequently, in SEGAN [130], an attention competition module focuses only on keywords by retaining their attention weights through a newly introduced attention regularization term, inspired from [271,272].

Attention at only the spatial level correlates words with partial regions, ignoring the feature selectivity of channels. As a result, spatial attention mainly focuses on color information while channel-wise concentration semantically associates significant parts with relevant words. Viewing that, ControlGAN [131] proposed a word-level spatial and channel-wise attention-driven generator generating coarse-to-fine images with a word-level discriminator. Furthermore, a perceptual loss is also adopted to reduce the randomness in the generation.

In a more current setting, an efficient, lightweight model, called TIME [135], is proposed that jointly learns a generator with an image-captioning discriminator. Since previous methods assess T2I as a uni-directional task, needing a pre-trained language model for text-image consistency, TIME neglects extra pre-trained modules. For this, transformers modeling cross-modal connections between image features and word-embeddings with annealing conditional hinge loss are devised, balancing adversarial learning. This model is a unified framework for T2I and image-to-text (I2T). The authors regarded attention in AttnGAN as a simplified version of the transformer, where features are flattened from a three-dimensional to a two-dimensional sequence. So, a 2D positional encoding for better attention operation is shown, which does not need sentence-level text features.

Similar to [129], Dynamic Aspect-awarE GAN (DAE-GAN) [136] refers to the importance of aspect in the input text. The model represents text information from multiple granularities of sentence-level, word-level, and aspect-level, for which, besides other attention mechanisms, the aspect-aware dynamic re-drawer (ADR) module is employed. ADR module contains two alternating components, the Attended Global refinement (AGR) module utilizing word-level embeddings for image enhancement and the Aspect-aware Local refinement (ALR) module for enriching aspect-level image details.

**Siamese Architectures:** Siamese architecture benefits from a small training dataset by learning more than one identical subnetworks in parallel, having shared parameters operating on a pair of inputs. The goal is to learn a similarity function for grouping inputs with similar patterns.

Above, we mentioned SEGAN [130] as the model with the attention competition module. This model adopts the Siamese architecture for semantic alignment through ground truth images by minimizing the feature distance between the generated and original image while maximizing for another image with a different caption. Motivated by Focal loss [273], sliding loss is applied to adapt the relative importance of easy and hard samples.

Text-Segan [137] highlights the importance of controlled negative sampling to improve GAN training, demonstrated on cGAN. Rather than selecting random mismatching negative samples for learning, negative samples are picked based on semantic distance from positive class examples, following curriculum learning [274]. Moreover, the auxiliary classification task for T2I can cause a decrease in diversity, so a regression task for semantic correctness based on the semantic distance to encoded text is employed.

SDGAN [138] also employs a Siamese architecture with two branches, individually processing text to produce an image from shared parameters. Similar to SEGAN, feature distances are minimized and maximized depending on whether there is an intra-class pair (captions from the same image) or inter-class pair (captions from different images) by the use of contrastive loss [275]. As a result, semantic commons are learned with a possibility to skip fine-grained semantic diversity, requiring a new module of semantic-conditioned batch normalization to adjust visual feature maps from textual cues.

**Knowledge Distillation:** Knowledge distillation is a transfer learning method by transferring knowledge from a teacher model to a student model, initially proposed for model compression [76].

Introducing knowledge distillation in GAN is first explored by KTGAN [139]. This study introduced two mechanisms for fine-grained T2I. First is the alternate attention-transfer mechanism (AATM), which alternatively updates the word and image sub-region attention weights. The second one is the semantic distillation mechanism (SDM), where a trained image-to-image encoder guides the learning of a text encoder in the text-to-image task.

T2I from multistage coarse-to-fine generation lack interactions among stages and ignores cross-sample consistency. So, ICSDGAN [140] proposed interstage cross-sample similarity distillation model based on GAN. This model uses cross-sample similarity distillation blocks in a three-stage network, where knowledge distillation is achieved from the refined to coarse stage.

**Cycle Consistency:** Models which form a cyclic process for learning a T2I generator, either with an image captioning (I2T) or an image encoder network (I2I), are classified as cycle consistency approaches. Nguyen et al. [276] showed a way to synthesize novel images through gradient ascent in the latent space of the generator network, maximizing activations of multiple neurons in a classifier network. In expansion, Nguyen et al. [141] introduced an additional prior on the latent code to improve sample quality and diversity. Furthermore, a unified probabilistic interpretation of activation maximization methods is provided, called Plug and Play Networks, which comprises a generator and a replaceable condition network. This condition network can be a classifier or a captioning network, where the goal is to iteratively find a latent code for the generator that maximizes a feature activation in the feedback network. Among the proposed variants of PPGN models, the Noiseless Joint PPGN model comprising a GAN and three interleaved denoising autoencoders (DAE) gave the best performance.

Hao et al. [142] gave a primitive cycle consistency approach for T2I by training Image–Text–Image (I2T2I), which integrates two separate models for improving T2I. Deep CNN-RNN for image captioning and image–text mapping added with the GAN-CLS module build I2T2I.

As an inspiration from CycleGAN [190], MirrorGAN [143] generates images by re-description architecture through learning semantically matching representations between images and text. It is accomplished by appending a captioning network to generate a semantically similar caption of the synthesized image with the original input. Sentence and word embeddings for global and local attention, respectively, guide the cascaded generator, which is in line with an image captioning network [277] for producing a caption of the newly generated image that is made consistent with original input text by cross-entropy-based reconstruction loss.

SuperGAN [144] is similar to MirrorGAN in terms of cycle-consistent adversarial learning with a cycle-consistent loss. However, its authors proposed a new evaluation metric for measuring sample diversity, and instead of [277] as a captioning model, they trained a CNN-RNN model from AlexNet and LSTM.

Lao et al. [145] learned to disentangle style via noise and content via text in the latent space of a GAN in an unsupervised manner, motivated by adversarial inference methods [278,279]. They used a supplementary encoder that infers the two latent variables, where the cycle-consistency loss retains consistency between the encoder and decoder. Added to the adversarial loss, a discriminator helps to distinguish between joint pairs of images and latent codes.

**Memory Networks:** Networks that harness the information from explicit memory storage with attention can be organized into a distinct category, so we cluster T2I GAN models which employ memory structure.

Most of the existing GAN-based methods for T2I generate images in a coarse-to-fine manner, which is highly dependent on the quality of the original image, where a fixed text representation for image refinement further worsens the result. Therefore, DM-GAN [146] utilizes a dynamic memory module to refine initial fuzzy images with a memory writing gate to select important text information from initially generated images. Additionally, to adaptively fuse the memory and the image features information, DM-GAN uses a response gate. This model operates on unconditional adversarial image and text-conditioned image–text matching loss.

Different from DM-GAN, CPGAN [148] designed a memory structure that analyzes the textual content during text encoding by examining the semantic correspondence between all vocabulary words with visual contexts across relevant images. Meanwhile, the images are generated in an object-aware manner with the help of a conditional discriminator for fine-grained correlation between words and image sub-regions. In summary, three components perform content parsing: Memory-attended text encoder, object-aware image encoder, and fine-grained conditional discriminator for text–image alignment.

**Contrastive Learning:** A popular form of self-supervised learning is contrastive learning. It encourages augmentations (views) of identical input to have a close relationship than the augmentations of different inputs [280]. Thus, studies that exploit this technique are mentioned under this topic.

Recently, synthetic images which are more coherent, clear, and photo-realistic are modeled from XMC-GAN [151] via multiple contrastive losses, which capture inter-modality and intra-modality correspondences. This model uses a simple one-stage GAN with an attentional self-modulation generator enforcing text–image resemblance with a contrastive discriminator as critic and feature encoder for contrastive learning.

Since human-annotated captions have significant variance, the linguistic discrepancy between captions causes deviating images. Consequently, Hui et al. [150] developed a contrastive learning approach for semantically consistent visual and textual representations, where consistency for synthetic images is enhanced during GAN training. Because their technique is flexible and can be implemented to any existing GAN model, AttnGAN and DM-GAN are set as the base methods. In contrast to XMC-GAN, the authors implied different objectives for contrastive loss, among caption–caption pair and fake–fake pair, which are complementary to the ones in XMC-GAN.

**Unconditional Models:** Unconditional image generation is promising and comparatively easier than the conditional task because of the uni-modality. Lately, the progress in this domain has encouraged researchers to adapt the architecture of these unconditional models for T2I.

Similar to [281], a progressively growing generator and discriminator during training is employed in Bridge-GAN [152]. It uses an intermediate network, following [157], to map text embedding and noise into a transitional space acting as a bridge with two mutual information-based losses to enhance reality and consistency. The mutual information objective optimizes the transitional space and improves quality, aimed at learning interpretable representation to reduce the cross-modal discrepancy.

BiGAN [278] has shown interesting results on class-conditioned image generation, adapting which Douglas et al. [153] presented T2I. Unlike conditioning augmentation (CA) in StackGAN, which uses the normal distribution to smoothen the data manifold, they introduced sentence interpolation (SI) as a deterministic function that can create interpolated sentence embeddings from all captions per image.

An extension to StyleGAN [157], the same authors proposed textStyleGAN [155] to generate higher-resolution images with image manipulation option. A pre-trained image–text matching network [156] computes word embeddings concatenated with sentence embeddings and noise to obtain a linear mapping for producing intermediate latent space. Moreover, an attentionally guided generator with a modified discriminator having two additional losses is used. These two losses of cross-modal projection matching (CMPM) and cross-modal projection classification (CMPC) losses [156] aid in aligning input text with image. As well as generation, image manipulation is possible by finding directions in the latent space correlating to different attributes.

Robin et al. [158] proposed a network-to-network (N2N) model for unconditional T2I. They train an invertible network [282,283] to fuse the pre-trained BERT and BiGAN model while translating their representations for T2I. The most significant contribution is the domain transfer which can help reuse expert models without learning or fine-tuning them.

Similar to N2N, the authors of FuseDream [159] showed a training-free, zero-shot, and customizable technique for T2I. Instead of BERT, they utilized CLIP [224] for text, whereas image generation is again from a BiGAN latent space. However, this fusion of two models is not an easy task, so with the help of three new techniques, the CLIP score is optimized in the GAN space. Among the three techniques, the AugCLIP score robustifies the standard CLIP score, over-parameterization optimization enables navigation in the non-convex GAN space, and composed generation with bi-level optimization generate multiple images to overcome data bias.

In the last few months, LAFITE [160] explored the latent space of the CLIP model for T2I without the use of text-annotated image data. This requirement of text-conditioning is relieved via generating text features from image features, considered a language-free model. Contrastive to the above models, this study utilizes StyleGAN2 [284] for latent space of image features.

#### 4.4.2. Supervised T2I

Due to the enormous research for T2I with GAN, exploration is not only limited to GAN models. Instead, various studies have shown the use of additional supervision to enhance the consistency and quality of the images. Generally, models with more than one supervision are better, with added annotation for the training data as a trade-off. Hence, after direct T2I, we review supervised methods which use extra annotation along with the text. More clearly, multiple captions, instead of one, for better textual consistency, dialogues for T2I as interactive methods, image layouts for controlled generation, scene graphs for better image understanding, and semantic masks for high-quality images are the mentioned extra supervision annotations for T2I.

**Multi-captions:** Text and image domains have a large dimensionality gap, causing insufficient information from a single caption. So, we cite the models that signify the importance of multiple captions for T2I.

Many existing methods ignore the use of multiple captions, where a single caption is limited and hardly contains the image concepts. C4Synth [162] addressed this by proposing a new cross-caption cycle-consistency model and a recurrent variant of it, inspired by CycleGAN [190]. The model follows a consistent hierarchy of text–image–text by predicting the caption from the generated image and matching it with the succeeding caption from multiple captions. However, this model is limited by the number of input captions, so a recurrent variant removes this limitation, called recurrent C4Synth.

Another approach that makes use of multiple sentences is GILT [163]. Unlike C4Synth, this model generates an image from a long text that does not explicitly mention its contents. The model is experimented with StackGAN++ on the Recipie1M [164] dataset, having cuisine images with corresponding ingredients and instructions as textual data.

Different from C4Synth, which requires many inferences for image generation with an additional captioning model, RifeGAN [165] directly generates an image once per execution and without the need of a captioning model. This function is due to enriching the given caption from prior knowledge from the training dataset and a caption-matching method by using an attentional text-matching model called self-attentional embedding mixture (SAEM).

Studies relating to semantic consistency among text and images overlook the semantic correlation between related texts as described in MA-GAN [167]. This method utilizes a single-sentence generation and multi-sentence discrimination (SGMD) module with a progressive negative sample selection mechanism (PNSS) to mine suitable negative samples for better training.

**Dialog:** Dialogue in a real-world scenario aids the drawer in rectifying and improving an image through feedback. Unlike multiple captions, dialogue conditioning focuses on the interactive generation, where each pair of query–response correspond to an intermediate result.

ChatPainter [168] is an excellent example of the model which leverages dialogue from the dialogue dataset [285] besides captions to generate images, for which Skip-thought provides embeddings. StackGAN, meanwhile, is employed for image generation.

The authors of GeNeVA [169] introduced a task named generative neural visual artist. This task involves a conversation between a teller and a drawer by adopting a recurrent GAN architecture for iteratively modifying the images. Because of this novelty, they created the i-CLEVR dataset, which is a sequential version of CLEVR [286] with text descriptions. Furthermore, a relationship similarity metric is presented to evaluate the positioning of objects by the model.

When dealing with dialogue for T2I, during training, there is a need for supervision at each turn. Moreover, it is challenging to evaluate the consistency between dialogues and images. Therefore, VAQ-GAN [171] showed that QAs are better than dialogues in this manner. Built on AttnGAN-OP [175] it has three key components, QA-encoder, QA-conditioned GAN, and an external VQA loss using VQA model [287] utilizing the VQA 2.0 dataset [288] with additional layout supervision. This study considers VQA model accuracies for evaluation between input QA and image.

SeqAttnGAN [172] is proposed for image manipulation uses multi-turn commands and is a form of interactive image generation. Since interactive image editing for fashion is new, two new datasets, Zap-Seq and DeepFashion-Seq, are also presented in this study.

Like VQA-GAN, VQA-T2I [173] use VQA data but without modifying the architecture to be effectivly applied to any model. A simple concatenation of QA pairs with other annotated data for training and an external VQA loss can significantly improve the results for T2I.

**Layout:** Layout-to-image generation [289,290,291] is captivating research where an image is drawn from objects defined by bounding boxes and labels. It ensures better-localized objects, which is user-controlled. Naturally, combining layout with text for T2I is explored by some studies.

GAWWN [43] is one study that can control object location and pose for T2I through bounding boxes or keypoints. The text encoder used in this study considers the average of 4-captions. For bounding boxes, noise and text embedding is concatenated to feed the generator, having local and global path. In keypoint annotations, for the location representation of various object parts, the model is adjusted with a necessary consideration of keypoint conditioning. It is worth mentioning that it is highly unlikely that the user might specify all keypoints in the description.

Comparable to GAWWN, Hinz et al. [175] also suggested the use of layout for T2I, but without the use of a detailed semantic layout. So, from the given bounding boxes and labels, they initially generate an intermediate layout for image generation. The model utilizes StackGAN and AttnGAN with considerable changes. The generator and discriminator consist of two streams, the global pathway and the object pathway.

As for AttnGAN as baseline architecture, OP-GAN [177] modified it for object-centric image generation with multiple object and global pathways, similar to [175]. Besides this model, a new T2I evaluation metric, named semantic object accuracy (SOA), is suggested in this study.

The model in OC-GAN [178] defines a scene-graph-based retrieval module (SGSM) to improve layout fidelity, with conditioning on instance boundaries for generating sharp objects. This model generates images from the layout, where the layouts are obtained from scene graphs. Further, SceneFID is proposed for a multi-object dataset as an evaluation metric.

**Semantic maps:** Semantic maps are different from layouts as they provide a more precise object shape, whereas image layout only provides bounding boxes with labels. Thus, we group studies which use semantic maps or masks for text-conditioned image generation.

Following a two-step generation, text-to-semantic layout from a layout generator and layout-to-image from an image generator, Hong et al. [180] proposed a hierarchical approach for T2I. The newly designed layout generator constructs a semantic layout in a coarse-to-fine manner by generating bounding boxes for objects and then refining them to estimate the object shape inside.

Identical to [180], Obj-GAN [184] also generates in two-step process. However, it consists of an object-driven attentive generator with an object-wise discriminator. This generator uses GloVe [170] embeddings of object class labels to query relevant words in the sentence, whereas the discriminator implements a Fast R-CNN [292] to provide feedback about object realism with matching layout and text.

To mimic the human strategy for T2I, LeicaGAN [186] decomposed T2I into three sequential phases, learning multiple priors, imagination, and creation. For the first phase, the text–image encoder and a text–mask encoder learn semantic and layout priors. The second phase combines these priors with added noise to stimulate the imagination. Finally, a cascaded attentive generator with local and global features successively generates an image.

Naturally, images are not provided with semantic masks and model-generated semantic segmentation maps are often noisy without instance information. With this in mind, a work by Pavllo et al. [187] on exploiting weakly supervised sparse mask setting, combining detailed mask with instance information, compared their model to the human-annotated mask, and semantic segmentation maps ensure localized image manipulation. In contrast to dense pixel-based masks, sparse instance masks can easily edit images by decomposing into the background and foreground.

By injecting image contours into the generative network, AGAN-CL [189] enhanced images generated from text. The model is trained to produce masks and consists of two sub-networks: a contextual network to generate image contours and a cycle transformation autoencoder for converting them to images. Moreover, the modified objective function includes perceptual loss, contextual loss, and cycle-consistent loss.

The authors of [191] introduced an end-to-end framework with spatial constraints from semantic layout for T2I. Adopting a coarse-to-fine image generation, they fused multi-scale semantic layouts with text and hidden visual features. During training, the generator produces an image and a corresponding layout for the relevant discriminator to distinguish between matching and mismatching the layout–text pair and real–fake layout pair besides the matching-aware task as in GAN-INT-CLS.

**Scene graphs:** Structured text (also referred to as scene graphs) for image generation is a promising approach for T2I. Unlike naturally existing static text, with intricate object interactions and concepts, scene graphs explicitly structure text as directed graphs, where nodes define objects and edge their relation. A vastly used dataset for image generation, MS-COCO, lacks scene graph annotation and is constructed from object locations [192]. However, more advanced data, such as visual genome [197], provide an average of 21 pairwise relationships per image.

The leading work of Justin et al. [192] utilized a graph neural network [293] for processing scene graphs [195] to predict an image layout containing bounding boxes and segmentation masks for every object, compared with ground truth during training. Then, a cascaded refinement network [193] subsequently generates an image from the combination of bounding boxes and masks.

An extension to [192] is given by Mittal et al. [194]. They proposed an interactive framework for incrementally growing scene graph-to-image generation through recurrent architecture. For image generation, changing the scene graph while preserving previous content allows a refined image to be updated and produced. For preserving the content of the previous image, the previous image is passed to the cascaded refinement generator, instead of noise, with the perceptual loss for image consistency.

The study from Seq-SG2SL [196] focused on the subtask of [192], semantic layout prediction and explored the non-sequential processing in a sequence-to-sequence manner. In this work, the scene graph is decomposed into a sequence of semantic fragments (SF) per relation, where the layout is the consequence of a series of brick-action code segments (BACS). As the two terms correspond to two unique vocabularies, a transformer-based seq-to-seq model plays the role of translator. Furthermore, a new metric named semantic layout evaluation understudy (SELU) is devised to assess the layout prediction technique.

Distinguished from [192], Oron et al. [198] separate the layout and appearance embedding with additional location attributes and stochasticity before creating the masks. Moreover, three discriminators for mask, object, and image are employed with perceptual loss. In this way, their work can achieve more control over image generation of higher-quality complex scenes.

Previous studies guaranteed image-level semantic consistency but lacked manipulation of every object. Accordingly, PasteGAN [199] introduced a semi-parametric method for image generation with scene graphs and image crops. For more appealing interactions in the final image, a crop-refining network and an Object–image fuser embed objects and their relations into one map to feed the image decoder. Although the above two networks operate to align the cropped images, selecting the most-compatible crop is addressed by the proposed crop selector.

Duc et al. [200] uses scene graph to predict initial object bounding boxes, from which they anticipate two-box relation units for each individual subject–predicate–object relation. After prediction, a convolutional LSTM [182] unifies all relation-units and converts them into a visual-relation layout because each entity is capable of having multiple relations. This layout reflects the scene graph structure and is used in a conditional pyramid GAN to generate images.

Another approach that uses scene graph for text-visual relation is VICTR [201]. The authors of VICTR proposed a new visual contextual text representation for the text–visual multimodal task, composed of five modules. First, the conversion of raw text to scene graph from scene graph parser is sent to GCN, having graph and positional embeddings, to form visual semantic embedding. This embedding, along with word attention, is changed to a visual contextual text representation. Finally, the encoded text containing words and sentences aggregates to generate visual contextual word and sentence representation. The joint representation learned is applied to the T2I task.

**Mouse traces:** A study with a novel annotation describing the text–visual relation is highlighted in [294]. It is unique to others as there is no explicit segmentation, just the rough markings, called mouse traces, with descriptive voice and text descriptions forming a Local Narrative dataset.

From the initial direction in [294], TRECS [203] uses its dataset, especially mouse trace annotations with detailed descriptions, to retrieve semantic masks for image generation. The mouse traces provide sparse, fine-grained visual grounding for the corresponding text defining an image.

**Manipulation:** For completion, studies for T2I under GAN-based models also explored editing the contents of an already given image. In contrast to generation, where only text is necessary, manipulation requires two inputs, the text and a given image to modify. Comparatively, manipulation is an advanced form of generation, where besides understanding text, learning image semantics is compulsory to know the exact location of modification. Currently, image manipulation from GAN models is studied under different variations, from global [226] to local [213,215,221], directly from text [208,210,211,214,216] or with additional supervision [218,219,222,225], and from the latent space of GAN models [223,229].

The first study to purely explore image manipulation from GAN was by Dong et al. [208]. They used a conditional GAN, following [7], where the generator encodes the input image to features and concatenates it with text semantics to decode the combined representation. Then the discriminator is allowed to distinguish the synthesized image which matches the target description.

A parallel work to edit the images globally is presented by Wang et al. [226]. They showed three trainable models based on RNN and GAN, having the same discriminator with a different generator that handles the text information differently. Namely, these models are the handcrafted bucket model, an end-to-end model, and a Filter-bank model. The generator possesses an encoder–decoder architecture with an RNN network. In addition, for the filter-bank as a general model, RNN is replaced with Graph RNN to prove its effectiveness. However, to evaluate the task, the lack of a suitable dataset encouraged the authors to collect a new dataset.

A limitation of SISGAN [208] is the use of a sentence-level conditional discriminator, which provides coarse training feedback insufficient to disentangle different image regions. As a result, TAGAN [210] proposed to split a single sentence-level discriminator into several word-level local discriminators. In this way, they can pay attention to specific visual attributes.

The authors of MC-GAN [218] proposed multi-modal conditioned image manipulation that uses a base image, text, and mask, to synthesize a foreground object on a background image. This multi-conditioning is due to a synthesis block that disentangles the foreground from the background in the training stage. This study is unique from SISGAN, as it manipulates the given image by creating an object on it rather than modifying the attributes of the original image.

In FiLMedGAN [211], the model is trained to manipulate the image for fashion. It uses feature-wise linear modulation (FiLM) [212] to relate and transform visual features from natural language, implemented in a modified version of SISGAN.

Limited research on image manipulation suffered from two problems: improper attention to specific parts of the image and low-resolution image generation. Therefore, Two-sided Attentive conditional GAN (TEA-cGAN) [213] proposed an attention mechanism on the generator, inspired by AttnGAN, with a discriminator following TAGAN. The two variants of the generator, single-scale and multi-scale, allow image manipulation at a single CNN layer or multiple layers. This multi-scale generator can produce high-resolution images.

Human visual appearance manipulation through natural language is rarely studied. Motivated by this, Text-guided Person Image synthesis (TGPIS) [219] investigated language-based human image manipulation task for user-friendly image editing. A two-stage framework is presented utilizing a GAN-based pose inference network with attention upsampling modules and a multi-modal loss for establishing semantic relations among images, poses, and text descriptions. In the first stage, a text-guided pose generator infers the pose, following [295]. The next stage obtains the target pose to transfer the text-based visual attributes to the reference image. Since it is dealing with three different modalities, a newly posed attentional upsampling (AU) module helps incorporate text-to-visual attention features with pose features at multiple scales. Furthermore, a new evaluation metric, VQA perceptual score, identifies the correctness of attribute change corresponding to the body part.

LBIE task from cGAN is explored in [214]. The authors highlighted the limitation of cGAN as it cannot learn the second-order correlation between two conditioning variables. Thus, they proposed a bilinear residual layer as an improved conditional layer to learn powerful representations based on SISGAN.

The direct concatenation of image and global sentence features along channel direction is responsible for poor performance in [208,210]. So, [215] devised another network called ManiGAN based on [131], having multiple generator–discriminator pairs along with two key components, namely affine combination module (ACM) and detail correction module (DCM). Utilizing this ACM module, they could only manipulate the image corresponding with the given text description. Apart from the new modules, they suggested a new evaluation metric to compare their results with those from previous methods. However, this metric seemed biased, so new techniques avoided this metric.

Previously, there was some trade-off between model size and image quality. So, a slightly different work [216] explored the idea of unsupervised learning, pointing to another yet undermined approach, text commands for image manipulation. Instead of using human-annotated data or complete attribute descriptions to learn the semantical alignment of text and image features, only a text command specifying the change is sufficient for image manipulation, given disentangled content features and attribute representations. Despite the simplicity of the text command still, there is much ambiguity. Consequently, their overall model utilized GAN with three separate encoders for content, attribute, and text to process the image before passing it to the generator. Based on the assumption that content and attributes are separable, GMM-unit [217] modeled the latter, while for non-deterministic translation, they combined various loss functions, including reconstruction, domain, adversarial, and attribute loss.

ManiGAN [215] consumed a lot of memory and training time but produced detailed images, so ref. [221] proposed a lightweight network composed of a single generator and discriminator and, thus, a reduced number of parameters as a trade-off for a slightly degraded-quality image. Furthermore, LWGAN addressed the limitations of the previous discriminators used in [131,210,215]. Therefore, a new word-level discriminator was introduced, which minimized the cross-entropy between word-weighted image features and target labels, obtained by labeling each word. Comparatively, two image encoders, Inception-v3 and VGG-16, were used to obtain the semantic and detailed image, respectively. However, the text encoder was the same as the previous studies, bidirectional RNN. In contrast, for text smoothing and text–image feature concatenation, conditioning augmentation (CA) and text–image affine combination module (ACM) were adopted, respectively.

Sometimes images are distorted and comprise incomplete regions. Studies that focus on filling the missing part of an image are termed neural image inpainting. A similar study, named TDANet [222], proposed an inpainting model harnessing the text information. First, the model uses a dual-attention mechanism to extract corrupted region semantic information by comparing text and image areas through reciprocal attention. Next, an image–text matching loss maximizes the semantic similarity between the text and image.

Advancements in GANs can generate high-quality photorealistic images, specifically from StyleGAN. An inspiration to learn the latent code of this network for image manipulation, ignoring other modalities. Moreover, the models for T2I are mostly limited to a single task, either generation or manipulation. From these issues, TediGAN [229] proposed a novel unified framework for both multi-modal image generation and manipulation to create diverse high-resolution images without multi-stage processing. Additionally, a GAN inversion technique capable of mapping information to a common latent space of StyleGAN is suggested, harnessing knowledge from multi-modalities. The implementation of the TediGAN involved three key components, StyleGAN inversion module, visual-linguistic similarity module, and instance-level optimization. For practical evaluation of this model, focusing on T2I for faces, a new dataset is introduced, named Multi-modal Celeb-HQ.

Similar to TediGAN, StyleClip [223] explored the best available vision–language joint representation model, CLIP [224], for text-based image manipulation by learning StyleGAN [284] latent space. Additionally, three combination techniques, latent optimization, latent mapper, and global directions, are also analyzed to investigate the benefit of combining these two models. The first two methods work in W+ space, where the former optimizes this space of a given image by minimizing CLIP-space loss for each image–text pair. In contrast to other similar models, such as DALLE and TediGAN, this model requires less computational power, and the quality of the generated output is improved.

Togo et al. [225] exploited style-transfer-based image manipulation framework. Their framework has three components, image captioning, style image generation, and style transfer net. They can perform image manipulation without the style image, and follows a module-based generative model.

**3D scenes**: Some studies in T2I explore deep generative models, especially GAN, for creating 3D scenes from the given text. However, due to the limited research in this field, the generated results are far from the real-world scenes and mostly rely on retrieval-based tasks [296,297,298].

Motivated by the limitations of retrieval-based 3D scene generation, Text2Shape [120] proposed an end-to-end instance-level association learning framework for cross-modal associations between text and 3D shapes. First, it learns a joint embedding, inspired by [299], of text and 3D shapes for the text-to-shape retrieval task, then introduces a text-to-colored voxel generation task with conditional Wasserstein GAN, following [300]. For the new technique, two new datasets are shown to be effective for evaluation. This model is different from GAN-INT-CLS as it does not require a pre-trained model or massive annotated data for training.

In the previous method [120], generating high-resolution 3D shapes requires extensive GPU memory or a long training time. So, Fukamizu et al. [121] considered the low-resolution problem and followed a two-stage approach by using StackGAN knowledge.

A different application of a text-conditioned deep generative model for the 3D scene is shown by Chen et al. [122]. They applied the knowledge of Graph scene parser [123] to obtain the layout by a graph-conditioned layout prediction network (GC-LPN) with language-conditioned texture GAN (LCT-GAN) to generate 3D models of houses. The overall task is split into building a layout and refining with texture synthesis. As a challenge to the proposed application, no dataset exists in the literature, so they introduced a new dataset called the text-to-3D house model.

## 5. Story (Consistent)

After mentioning the advances in the text-to-image domain, we shed light on the studies which focus on generating visual stories from the given natural language. In contrast to T2I, text-to-story (T2S) is one step ahead, where the generated images are coherently consistent with the previous scene based on the semantics but without any continuity in the generated frames. This is different from video since it lacks continuous frame prediction, having a temporal relation to show a smooth motion transition. However, the literature reports only a few studies on the story-generation task, most of which are retrieval-based [301,302,303], while some pay attention to GAN models [31,231,233,235,237,239] and almost none of the other generative models are explored, except [230].

### 5.1. GAN Model

Distinct from the story-retrieval task [303], GAN-based models implement the generation of an unseen image rather than finding the best match for the given text. Limited research on the story-visualization task from the text typically focuses on GAN models.

The first-ever implementation of generating visual representations of textual stories by a GAN model is studied in StoryGAN [231]. The authors named this task story visualization, and from the multi-sentence paragraph, they visualize the story by a sequence of images per sentence. The model consists of a deep context encoder to track the story and two discriminators for image quality and story consistency. StoryGAN follows a two-level GAN framework with RNN to incorporate the previous image with the currently generating image supervised by a context encoder module. This module contains a stack of GRU and Text2GIST cells. Additionally, two new datasets, called Pororo-SV and CLEVR-SV, are collected for the newly introduced task.

To further improve the visual quality and semantic relevance, PororoGAN [31] jointly considers story-to-image sequence, sentence-to-image, and word-to-image patch alignment. Precisely, they introduced an aligned sentence encoder (ASE) to improve global relevance and an attentional word encoder (AWE) for local consistency. Besides previous discriminators, image patch discriminator is added to enhance the image reality.

Improved-StoryGAN [233] is an extension to StoryGAN. In this work, simple convolution is replaced with dilated-convolution, inspired by [234], to expand the receptive field of the kernel. Additionally, the weighted activation degree (WAD) introduced in the discriminators enhances consistency between images and the target story. Finally, the use of gated convolution in initial state encoder obtains better feature representations with Bi-GRU as context encoder.

Emphasis on preserving the global consistency of characters and scenes across different story pictures, in CP-CSV [235], a character-preserving coherent model, is shown, which uses a segmentation mask to separate the foreground from the background. The framework is split into three crucial modules: story and context encoder for feature representation learning; figure–ground segmentation as an auxiliary task for preserving characters; and figure–ground generation to generate a sequence of images. Moreover, the authors of CP-CSV suggested Frechet Story Distance (FSD) as an evaluation metric for this task.

Since limited text describing an image in the story lacks semantic alignment, DUCO-StoryGAN [237] implemented dual learning via video redescription. This dual learning with a copy transform mechanism in the GAN framework enables sequentially consistent stories. Furthermore, to model the correlation between word phrases and corresponding image regions, a memory-augmented recurrent transformer (MART) [238] is employed. However, the lack of proper evaluation metrics encouraged the authors to present a diverse set of new metrics.

The authors of DUCO-StoryGAN enhanced the story-visualization task in VLC-StoryGAN [239]. They showed that integrating linguistic information with common-sense knowledge, motivated by [304], can generate better results. From CP-CSV and DUCO-StoryGAN, which use segmentation mask and video captioning, respectively, as an auxiliary task, generate uni-modal outputs. Therefore, to combine the benefits of both, dense captioning as the dual task is applied. Moreover, implementing an extra intra-story contrastive loss between image regions and words improves semantic alignment between captions and visual stories.

### 5.2. Autoregressive Model

From one reported work on an autoregressive model for story visualization, multiple descriptions per image are essential for the generalization of the generator. However, the Pororo-SV dataset consists of only a single text–image pair, which the previous studies [31,233,235] are limited to use in training. Recently, an autoregressive model based on the transformer, called C-SMART [239], studies story visualization generated from text. The name C-SMART emphasizes the cyclic story visualization by a multi-modal recurrent transformer. The term cyclic refers to the image–text–image stream, where pseudo-text generated during this approach helps train a T2I generator. Furthermore, to achieve the temporal consistency among images, a dynamic gated-memory module is applied to the multi-modal recurrent autoregressive transformer following [237,238].

## 6. Video (Dynamic)

Video generation from text is a significant and challenging task. It shares some similarities with T2I and T2S as it generates new visual content as video frames from text conditions. However, the main difference between the other two is the continuity of the output, as video frames are temporally more consistent and should share consistency throughout the video. Initial research on text-to-video generation (T2V) utilizes rule-based retrieval models [305,306,307,308,309] that lack the power to create new videos and are limited to a set of pre-defined options. However, after the advent of T2I, a few studies attempted T2V using either autoregressive models, VAE, or GAN.

### 6.1. VAE Models

Models under VAE selectively learn by maximizing the variational lower bound of the observation while keeping the approximate posterior distribution close to the prior distribution. So, now we mention the models leveraging this generation technique for creating video frames, where frames are made consistent with the help of an RNN network.

Starting from Sync-DRAW [250], T2V is pioneered by the combination of a recurrent attention mechanism with VAE. The attention mechanism attends to each frame in synchronization, while VAE learns the latent distribution of the whole video at the global level. This work is similar to [6], but spatial attention differs from spatiotemporal attention.

From the authors of Sync-DRAW, an improvement for T2V in [252] suggests the use of captions combined with long-term and short-term dependencies between video frames for incrementally generating video. This way, they can perform variable length semantic video generation from unseen captions, maintaining a strong consistency between consecutive frames.

In parallel, a hybrid framework employing VAE and GAN for T2V is given by Li et al. [253] in the same year as Sync-DRAW. They propose to extract static and dynamic information from the text to train a conditional generative model. The static features, called gist, sketch text-conditioned background color, and object layout, where transforming text to image filter models better dynamic features. Additionally, it provides a method to construct a new training dataset from Youtube videos accompanied with titles and descriptions.

The need for high computational power limits T2V for generating compelling results, so GODIVA [254] trained a large-scale model capable of creating videos from the text in an autoregressive manner using a three-dimensional sparse attention mechanism. It is distinguished from GAN and utilizes the VQ-VAE approach while sharing similarities with autoregressive models. This pretrained model uses the HowTo100M [310] dataset containing more than 136 million text–video pairs to scale the generation for zero-shot settings. However, previously poor evaluation metrics led to the need for a new relative matching (RM) metric for quality and semantic match.

### 6.2. Auto-Regressive Models

Sequentially generating new data from the previous data is termed autoregressive. However, we consider some studies [243,244] autoregressive due to the sequential prediction of frames, similar to others, but without using GAN or VAE models. These models typically fuse the two domains, text and video, for learning joint embedding.

#### 6.2.1. Generation

In CRAFT [243], text-conditioned video creation is completed by a compositional retrieval task. Following the caption, the model sequentially predicts a temporal layout of objects and retrieves the Spatio-temporal entity segments from a video dataset, where the fused segments create the final video. Consisting of three parts, layout composer, entity retriever, and background retriever, the model first predicts the location and scale of an entity and then seeks the best entity with a suitable background. These components are sequentially trained on the newly proposed dataset of FLintStones. Precisely, this model is retreival-based T2V.

The unstable training in GAN and blurry videos from VAE initiated the need for a similar study, CMDL [244], where instead of GAN or VAE, a deep learning model utilizing a dual learning algorithm is proposed. The model learns the joint embedding using sentence-to-video and video-to-sentence to learn the bidirectional mapping between the two domains. It is realized with the help of a multi-scale text-to-visual feature encoder for global and local representations.

#### 6.2.2. Manipulation

For increasing the complexity, SA3D [247] introduced the proof of concept for 3D scene generation from text, which is different from previous works on 3D scene generation as it allows free-form text descriptions. Therefore, they showed a two-stage pipeline that can generate static and animated scenes using a transformer-based text encoder with a multi-head decoder for predicting object-specific features per head to create an abstract layout. This layout is passed to a scene renderer [249] to generate the final 3D scene or video. However, due to the research gap in this area, they created a synthetic dataset, called IScene, for experimentation.

### 6.3. GAN Models

As in other tasks, T2I and T2S, T2V is also studied more under GAN models than others. The reason for this is the well-established research of GAN for T2I, so extending it to T2V is natural. However, when dealing with the consistency of frames for video, a challenging task, several studies are found in the literature, among which only a few targeted T2V, mentioned in this paper.

#### 6.3.1. Generation

As previously mentioned, in [253], a combination of the GAN framework with conditional VAE (CVAE) explores T2V. Utilizing three components, a conditional gist generator for the intermediate step using CVAE and a video generator with a discriminator, they train an end-to-end model.

In the successive year, TGANs-C [255] proposed another framework to explore the semantic and temporal coherence in GAN for generating videos. Typically, the input noise concatenated with caption embedding is sent to the generator to transform into a frame sequence using 3D Spatio-temporal convolutions instead of 2D. Instead of a single naive discriminator, the model consists of three discriminators. The first one separates real from synthetic videos, the second aligns the frames with caption while discriminating between real and fake, and the last emphasizes motion smoothness across frames. The frame-level discriminator allows the establishing of a connection between the caption and frames, where the motion level is responsible for coherency over frames.

Previously, simple conditioning on text [250,255] or substituting 2D with 3D convolutions, as in [253,255], is not feasible as the 3D layers may have poor frame quality [311], while 2D layers fail to tackle temporal dependency. So, IRC-GAN [256] explicitly handled the two components of T2V generation, quality, and semantic consistency by integrating LSTM cells with 2D transconvolutional networks. In this way, the 2D transconvolutional layers focus on more details than 3D. However, to properly align the semantics of video and text, the inefficient simple matching between the two is added with mutual-information introspection for consistency. For this, a two-stage training process is adopted, where a seq2seq text encoder with an introspective network extracts the mutual information between the text and video in stage one, and stage two tries to minimize the distance between the two.

Text-filter conditioning GAN (TFGAN) [257] addresses the limitations of [253,255], which require 3D convolutional layers for fixed-length videos, trained on low-resolution data with simple text–video feature concatenation. Consequently, following [312], a shared frame generator employing a recurrent network in the latent space resolves the fixed-length video problem. Next, the use of ResNet-style architecture in GAN allows higher-resolution results. Furthermore, utilizing the new multi-scale discriminative convolutional-filter text-conditioning scheme enhances the text-video correlation. However, existing datasets are not suitable to validate the effectiveness, so a new synthetic dataset is proposed.

Since text-to-video generation is new, many earlier works deal with limited synthetic or real data. Hence, Mazaheri et al. [258] showed that instead of traditional RNN and deconvolutions, which add extra parameters and complexity, temporal dynamics can be captured by regressing the latent representation of the first and last frame from the text followed by a context-aware interpolation method for in-between frames. Afterward, to revert representations back to RGB frames, an upPooling stacking block is introduced that can progressively increase resolution. Additionally, their discriminator encodes videos on single and multiple frames for 2D and 3D CNN, respectively. As a result, they generated videos from free-form sentences on more challenging datasets of A2D [313] and UCF101 [314].

The authors of TiVGAN [259] make use of the well-studied T2I task to explore T2V and propose a text-to-image-to-video training framework using GAN. In the first step, a T2I model creates a high-quality single video frame conditioned on text, then gradually evolves to create longer frames with the given text. This step-by-step learning stabilizes the training while producing high-resolution video. However, for further stabilization, several other techniques are also introduced in this paper.

#### 6.3.2. Manipulation

Very recently, Fu et al. [260] introduced a language-based video editing (LBVE) task to semantically edit the content of the video given an input video, realizing video-to-video (V2V). They proposed a multi-modal multi-level transformer that dynamically learns the correspondence between video perception and language at different levels. Due to the newly defined task, they gathered three new datasets containing two diagnostic and one natural video with human-labeled text. The model consists of a 3D ResNet to encode the video frames, combined with the sentence, and word-level text embeddings are fed to the multi-modal multi-level transformer. Inside this, a multi-level fusion (MLF) mechanism performs the cross-modal fusion between text and video. Then, utilizing this fused representation, the frame generator produces a video that is discriminated by a dual discriminator, following [315].

## 7. Datasets

After listing the various T2Vo methods, now we present the list of the datasets found in these studies, as shown in Table 3. We classified these datasets into images and videos and added sections based on particular attributes and characteristics.

Many T2I papers adapt to three datasets, Oxford-102, CUB, and COCO, whereas other image datasets serve as either zero-shot learning, use of additional annotation, or for a different task. For T2S, the most used data is Pororo-SV, and in a rare case, another dataset is also used for generalizability. However, T2V follows diverse datasets, where KTH, MSVD, and Moving-MNIST are commonly seen in the literature.

In Table 3, we highlight the source of the dataset with given annotations, but in a few cases, additional annotations are added by other studies, which we marked by the reference of the paper in the annotation column. In almost all the datasets, data separation into training, validation, and testing are given by the publishers. However, where there is no clear distinction, different studies adopt different splits for evaluation.

Moreover, based on data collection, there are two types of datasets, real-world and synthetic. Real-world data is often obtained from the internet or cameras and is generally complex with a large storage capacity. On the other hand, the synthetic type is easy to create and usually requires less storage and computational resources.

## 8. Evaluation Metrics and Comparisons

To complete the discussion of this study, we enlist the evaluation metrics used for various T2Vo methods, split between automatic evaluation metrics and human-based. Despite the flaws of current automated metrics, we compile different evaluation metrics and their scores in a separate table, Table 4, to T2I because of detailed research in this domain, whereas another table, Table 5, is devoted to T2S and T2V.

### 8.1. Automatic

First, we discuss the automatic evaluation metrics used for T2Vo tasks, followed by human-based studies.

#### 8.1.1. T2I

Among the given automated metrics, there are two distinct divisions: one evaluates the quality of visual output and the other for measuring the semantic alignment between visual and textual data. Table 4, for automatic T2I evaluation metrics, lists only the frequently used metrics, such as Inception Score (IS), Fr´echet Inception Distance (FID), Structural Similarity Index (SSIM), and Learned Perceptual Image Patch Similarity (LPIPS) for quality. Metrics such as Semantic Object Accuracy (SOA), Visual-semantic Similarity (VSS), R-precision, and Captioning metrics help evaluate the image–text alignment.


**Quality metrics**


IS [116] is a numerical assessment method that computes a conditional label distribution by classifying generated images using a pretrained Inception-v3 network. This distribution should have low entropy to indicate the meaningful images from the generation network, showing diversity. However, it fails to capture the over-fitting problem and cannot measure intra-class variations [177].

FID [394] finds the distance between the actual and the generated images using extracted features from a pre-trained network, which is more consistent than IS. For FID, multidimensional Gaussian is assumed, which is not necessary every time. Moreover, FID suffers from high variance when per-class samples are low.

SSIM is another image quality assessment method based on perception to measure the similarity between images. It considers image degradation as a perceived change in structural information while incorporating important perceptual phenomena, including luminance masking and contrast masking terms.

LPIPS [395] is the L2 distance between features extracted from a deep learning model of two images, closely resembling human perception. So, the higher the distance, the greater the diversity, indicating a better generative model.


**Semantic metrics**


SOA [177] utilizes a pre-trained object detection network to infer the objects within an image from the given caption. This metric evaluates the individual areas or objects rather than the holistic image as in IS or FID while considering the captions.

VSS [126] metric measures the distance between the generated image and its caption using two models that embed images and captions, respectively, and then minimizes the cosine distance between matching image–caption pairs and vice versa for mismatching.

R-precision [34] on the other hand, performs the same action as VSS, where instead of a VSS score between a given image and caption, it performs a ranking of the similarity between the real caption and randomly sampled captions for a given generated image. So, both these metrics, VSS and R-precision, do not consider the quality of individual objects.

Captioning metrics [180] try to evaluate the T2I models by comparing the original captions with captions obtained from generated images using a pre-trained caption generator. Then these two captions are compared by standard language similarity metrics such as METEOR, CIDEr, and BLEU. The main problem with these metrics is the one-to-many mapping, as one caption is valid for many. So, they are sensitive to n-gram overlap, which is insufficient for two sentences to convey the same meaning.

#### 8.1.2. T2S and T2V

The Table 5 for T2S and T2V indicates a diverse range of metrics as there is no standard. So, next, we briefly define these metrics.

T2S models typically employ classification accuracy as an evaluation metric for the story characters and the image frame. So, a classifier is trained on images generated by the network, and then its performance is checked on the original test dataset used to train the generative network. For the case of character classification, a pre-trained Inception-v3 model identifies the character in the generated image.

Meanwhile, [235], following FID and FVD, proposed FSD that measures the consistency between frames. FVD [396] evaluates a sequence of generated images and adopts Inflated 3D ConvNet for video but requires a minimum of seven frames. So, FSD based on [397] as backbone calculates the Frechet distance.

In terms of videos, metrics such as Negative loglikelihood (NLL) as a reconstruction loss, CLIP-similarity, and Relative matching (RM) [254] evaluate the text–video semantic match and domain-independent generation quality, respectively. Furthermore, some studies used GAM [398] as an evaluation metric that can directly compare two generative models by engaging them against each other. Its limitation is the use of only GAN models. In the video editing task, video activation distance (VAD) as the mean L2 distance between video frames using ResNext is adopted.

### 8.2. Human Evaluation

Even though the need for automated evaluation metrics is crucial, their lack of consistency and reliability is a bottleneck to the proper assessment of the T2Vo tasks. So, many studies additionally performed a human-based evaluation to better judge the quality of the generated output. A typical setup is to create the output from many models and then present it to a group of people to rank what they perceive as best. This evaluation technique is prone to two severe types of mistakes, inconsistent methods, and human error during evaluation, as people have personal likings that are dependent on many factors. So, we skip to these metrics in the paper for any comparison.

However, another way of evaluating the models is known as ablation study, which differs in terms of methods but is used to validate the performance of the given method. Furthermore, the use of subjective results of the visual output is another human-based comparison method commonly applied in the T2Vo tasks, as shown in Figure 9, Figure 10 and Figure 11.

## 9. Applications

Currently, various practical applications of T2Vo exist for industrial and commercial use. From the T2I task, the current applications include generating compelling images from the given text, which can be viewed as cross-modal information retrieval. Moreover, from the literature [208,211,221], the T2I task has another application for interactive and iterative image manipulation, especially useful for fashion and daily photography for a non-technical person. Additionally, missing regions of an image can be filled with visually realistic content while keeping coherence with image inpainting [222] guided by the text, a crucial task for image restoration. As identified in [122], we can also automate the laborious 3D house modeling task. Furthermore, the application of modifying the human appearances, including poses and attributes from the natural language [219] is useful for surveillance systems.

The T2S task is more interesting than T2I as it is close to human imagination. Due to the limited research in this field, the prospective applications include visualization of educational materials, such as the water cycle in a science lesson, and assisting artists with web-comic creation.

Over the internet, an extensive amount of data in images and videos accompanied with text serves as communication among users. In particular, video search engines or movie databases such as YouTube or IMDb have textual descriptions or comments about the video that describe the theme of the video in a shorter way. When in the reverse direction, T2V is formulated, which can be helpful for creating animated movies or visual representations of some concept. Like T2S, T2V is an immature topic with lots of limitations and gaps, so currently, practical applications are under development for this task.

### Open-Source Tools

T2Vo tasks are performed by one of two methods: developing a model from scratch or improving an existing model. Two essential tools are required to complete the task for both methods. First, a deep learning environment is necessary for developing a model, such as MATLAB (https://www.mathworks.com, accessed on 30 May 2022), C++ (https://isocpp.org, accessed on 30 May 2022), or Python (https://www.python.org, accessed on 30 May 2022). However, MATLAB is proprietary closed-source software, and C++ is typically complex compared to other programming languages, whereas Python is an open-source, free software that is user-friendly and widely used by researchers for deep learning. On top of that, many open-source libraries and frameworks optimized for Deep Learning (DL) and Machine Learning (ML) are easily available for Python. Some of them are:Pytorch, TensorFlow, Keras, and Scikit-learn; for DL and ML;NumPy; for data analysis and high-performance scientific computing;OpenCV; for computer vision;NLTK, spaCy; for Natural Language Processing (NLP);SciPy; for advanced computing;Pandas; for general-purpose data analysis;Seaborn; for data visualization.

Second, open-source datasets are needed to train the models as these models are data-driven. So, open dataset aggregators which help developers and researchers find the suitable dataset for their task are briefly mentioned as:Kaggle (https://www.kaggle.com, accessed on 30 May 2022);Google Dataset Search (https://datasetsearch.research.google.com, accessed on 30 May 2022);UCI Machine Learning Repository (https://archive.ics.uci.edu/ml/index.php, accessed on 30 May 2022);OpenML (https://www.openml.org, accessed on 30 May 2022);DataHub (https://datahubproject.io, accessed on 30 May 2022);Papers with Code (https://paperswithcode.com, accessed on 30 May 2022);VisualData (https://visualdata.io, accessed on 30 May 2022).

Another efficient way to advance T2Vo tasks is by improving the limitations of the previous models, which requires understanding their readily available source codes, datasets, and standard evaluation criteria. Generally, an article written for a model is not enough to thoroughly understand the detailed working of that model. So, papers published in top-tier journals or conferences sometimes provide their source codes hosted on GitHub (https://github.com, accessed on 30 May 2022), along with open-source datasets either hosted on a local server or online hosting websites such as Google drive.

## 10. Existing Challenges

After examining the broad taxonomy employing different datasets over various evaluations, we highlight the most commonly experienced problems when dealing with T2Vo.

### 10.1. Data Limitations

**Limited Data:** One of the most prominent assets of deep learning is a clean and accurately annotated dataset. Since deep learning models are data-driven and, in contrast to their counterparts, model-based, they require a massive amount of data for better understanding. In theory, these models can take an infinite amount of data, but unfortunately, we are limited to only a short version of it. Among many reasons for the limited amount of data, two of the most crucial ones include costly creation and error-prone annotation pipeline.

**Costly creation:** Primarily, data creation is a three-layered sequential task. Initially, a perceptual goal is defined for which there is a demand for the dataset. Next, there is a need for a reliable and unbiased setup to collect the data from millions of available random and raw data. Lastly, after data collection, a reasonably accurate and dense amount of annotations are required for this data to be used in training for deep learning models. Following this approach, we can estimate the cost by determining the use of available technology and resources such as high-speed internet connections, enormous storage, intense processing power, and trustworthy human labor, therefore leading to an expensive procedure for data creation.

**Error-prone annotation:** Apart from the costly setup, another significant problem when dealing with enormous datasets is improper annotations. As human labor is more expensive than an autonomous procedure, large datasets often employ automated ways to annotate the data, which again is limited to the model capability. However, human annotations show improved quality but are usually prone to human error, and for massive datasets, it is cumbersome to identify such errors. Hence, the lack of proper data annotation also causes a hindrance in learning the best model for T2Vo.

### 10.2. High-Dimensionality

**Image to Video:** As is prominent in Figure 8, an extensive amount of research is devoted to generating images from text, mainly using GAN models. Consequently, a considerable research gap exists for high-dimensional visual output such as stories and videos, increasing complexity by adding a third dimension of consistency, focusing on the correlation between previously generated output. For this reason, a few datasets and evaluation metrics are proposed in the literature for these tasks. Another challenge to these sophisticated tasks is the increase in model complexity resulting in the need for more computing power.

**2D to 3D:** Not only is the current research limited to images mostly, but it also vastly ignores the dimensionality in terms of object representation, 2D or 3D. Therefore, as is evident from our proposed taxonomy, the existing research for text-to-3D output is mainly retrieval-based, with scarce attention given to GAN-based 3D image generation. One particular explanation for this limited study is the additional variables involved, increasing complexity beyond the computational power of existing generative models. Since the current generative models still struggle to generate a text-guided realistic 2D visual output depicting a complex scene. So, adding further complexity by increasing the variables is yet to be resolved.

### 10.3. Framework Limitations

**Model limitations:** Although present generative models, especially Auto-Regressive, VAE, and GAN, produce compelling results when trained on a large dataset utilizing advanced techniques, they individually suffer from diverse limitations. For energy-based models, sampling from data distribution is not straightforward and implies a Markov chain, where mixing is a time-consuming task. VAE, on the other hand, produces results that tend to be unrealistic and blurry. Moreover, other issues with this kind of model utilizing posterior approximation include under-estimation [399] and amortization gap [400]. However, various techniques have been proposed in the literature to resolve such issues [401]. In continuation to VAE, a more appealing approach capable enough to produce realistic and sharp results for generative modeling is GAN, eliminating the need for Markov chains. Unfortunately, these models suffer from four main problems, namely unstable training, sophisticated architecture, slow training speed, and mode collapse. In the case of autoregressive models, based on the chain rule of probability, data sampling is inherently sequential and causes the slow processing of high-dimensional data with the further condition of ordered decomposition.

**Limited exploration:** Restricted by the limits of different models, the most explored model over the last few years is GAN due to the edge of better results compared to others. As a result, this indicates a void in the study of other models targeted at the generation of text-to-visual output. Given the constraints of different models, the benefits offered can be combined to overcome some limitations and produce better results, as explored by very few studies. It also highlights the lack of interconnection between various generative models for a specified task.

**Hardware-capacity:** Another issue related to the framework is hardware capacity. When dealing with a complex generative model or a hybrid model of more than one type, employing a large dataset of high complexity can lead to an overflow. The reason for this is the complex gradient-based learning of the models, which has an upper bound represented by the over-fitting and gradient vanishing problem.

### 10.4. Misleading Evaluation

**Lack of standard:** The most challenging task for T2Vo has been, until now, a fair and reliable evaluation method. Many studies tried to propose one-for-all evaluation metrics for such tasks but failed to identify a practical and authentic one. Various metrics highlight different strengths of the model but lack stability and do not fulfil the criteria of being selected as a standard. Therefore, from Table 4 and Table 5, multiple quantitative evaluation metrics are proposed to evaluate a single model. However, it is still not feasible to establish these metrics as standard.

**Unrealistic Scores:** In terms of quantitative scores, many evaluation metrics, such as R-precision, IS, and CIDEr, provide scores that have already achieved the upper bound of their performance. However, in reality, the generated output is not even close to a natural result, causing deception while indicating the flaws of these metrics. An R-precision for models that is higher than the actual data is observed, possibly due to the use of the same text encoders for training and evaluation [177]. Similarly, IS is likely to be saturated and overfitted and might be resolved by a large batch size [184]. However, metrics such as FID, FVD, FSD, Visual similarity, and SOA show a near approximation of the human judgment by marking bad scores to the generated output compared to the real data.

**Inconsistent scores:** Because the current evaluation metrics are biased and unreliable, many papers reported inconsistent scores of the same model. Although understandably, the scores might change depending on the implementation, resolution, and amount of samples, some of them are hard to explain. Most of the time, this is because the evaluation method is not precisely clarified, has no code, and is susceptible to change on the cloud storage compared to the ones reported in the paper.

**Score variation due to Data:** As the commonly used metric for evaluation, IS and FID are trained using Inception-v3 on object-centric data, ImageNet, which causes problems for evaluating complex scenes, highlighted by [177]. Consequently, ref. [178] proposed SceneFID to apply FID on object crops.

## 11. Discussion and Future Directions

After the outlined taxonomy, indicating the state-of-the-art methods evaluated by various inadequate techniques and current challenges, this section is devoted to summarizing the current progress in the field of T2Vo. Added to that, assessing the progress under current challenges, we also discuss the future research directions.

### 11.1. Visual Tasks

The visual output in current technology varies from images to videos and from 2D to 4D as animations. Similarly, natural language can be simple labels to captions or question answers to dialogues based on complexity, whereas, from an attention viewpoint, paragraphs, sentences, words, characters, and symbols are different forms. Due to the cross-modality, where natural language is convenient for humans, a one-to-many problem exists when dealing with visual data, since the text can represent diverse visual representations. So, bridging this gap between the two modalities is not trivial, and many studies offer multiple improvements to the first text-to-visual task.

Thus, in current research using deep learning models, images are the most-studied topic as they offer less complexity than others, and the most-explored generative model is GAN. However, some studies also paid attention to other related visual tasks of stories, intermediate between image and video, and videos guided by the text, where they require an understanding of previously generated results for maintaining consistency.

In particular, generation is not the only T2Vo task. Instead, some studies focus on editing the contents of an already existing visual data and are termed manipulation. Therefore, besides text and input visual data, they sometimes require additional supervision as semantic masks or layouts.

**Complex Visual Output** One aspect of text-guided visual modeling is to obtain a joint representation of the two modalities. Although interest in T2I started from [94,95] since its exploration using GAN [4,7], researchers shifted their attention to it and overlooked the broader picture. They tried to improve the T2I generation mainly using GAN and adapted two variations, GAN exploration and data exploration. However, recently, studies such as [231,250,253] following the T2I task have been interested in exploring a more challenging domain, T2S and T2V. Unfortunately, these tasks are recent and lack enough research to produce compelling results, as in T2I.

More importantly, research in [120,247] showed that T2Vo for 2D can be extended to 3D, which is far more challenging due to the perception of an additional variable. Among many practical applications of this research, field analysis such as car accidents [306], military tactical planning [402], house design [122], and movie creation [259] are possible.

**Generation and Manipulation** Modifying the existing data on purpose is crucial to many real-world scenarios. The work on editing images [204,208] share similarities with T2I generation but differ by the input to the model. Research on GAN for image manipulation is natural, following the synthesis task. In practice, manipulation is either local [204,205] or global [226], depending on the user requirements. Unlike image manipulation, story and video manipulation are rarely studied, where only one attempt at video manipulation [260] is found in the literature, and no work on stories. Because of this pioneering work, fully supervised training without zero-shot learning is not a limitation for this task, compared to others.

### 11.2. Generative Models

Progress in AI from retrieval to deep learning models experienced an increased performance for various applications. Initial cross-modal deep learning models [92,93,95] originated with variants of the Boltzmann machine that were impractical for a large-scale model because of the overwhelming dependency on the Markov Chain [250].

The text-to-image task adequately started with a simple GAN model having one generator and one discriminator with additional conditioning on text using basic adversarial loss. Then, a multi-stage pipeline with several losses deliberately removed the low-resolution and low-quality issues in the simple model. Next, attention was paid to more meaningful terms covered in the semantics between the text and image. Still, the lack of multi-object generation of the previous models leads to the investigation of new architectures for further improvements, including Siamese, cycle consistency, and knowledge distillation. Instead of specially designed architectures, latent space exploitation of the best text and image models to adopt for unconditional T2I modeling is also examined. One significant drawback of GAN is unstable training, for which researchers turned to other models such as VAE and autoregressive models. In the case of VAE, the main limitation is the blurry result, which is often attributed to the limited expressiveness of the inference models, the injected noise, and imperfect element-wise criteria such as the squared error [403]. Although autoregressive models are more stable, they suffer from global consistency problems [105], high-computational cost [103], and slow sequential inference [96], especially in images because of the locality bias and convolutional networks that focus more on local correlations. Thus, various solutions have been proposed to resolve such issues, including pretrained models and compressing images with VQ-VAE tokens. Interestingly, some other models are also explored for T2I, such as Knowledge distillation [106,107,139,140] and show a great deal of improvement to restrictions offered by other generative models.

Leading from T2I, generating visual stories from sequential text is a relatively new domain of research [231], despite its initial conception in [303]. T2S is similar to T2I except for maintaining a global consistency among the generated images. So, to maintain the consistency, recurrent models are employed with GAN having variations in discriminators [31,231,233] or utilizing additional data, such as segmentation mask [235]. Others consider the semantic alignment between the story and images through a cyclic network of text–image–text as an auxiliary task [230,237,239].

Similar to T2S, another significant area is the text-guided video generation with principal contributions using GAN [253,258] and some variations in VAE [250,252] and retrieval-based models [243]. However, the main difference of this method compared to the first two is the dependency and temporal consistency between consistent frames of the video, hence the named dynamic.

First attempts in this domain used retrieval models [306,307] that are primarily limited to a fixed set of rules. However, recent progress in the retrieval task using deep learning models of transformer [247] and CNN [243,244] improved this to free-form text for video creation. Although the retrieval task with deep learning models is fascinating, the restriction of specified visual output restrains its practical use for real-world applications. Moreover, the need for a large amount of data to search is another drawback. So, recurrent VAE-based models [250,252] show an improvement over retrieval-based models by depicting a more diverse range of outputs. However, the output blurriness is one particular flaw [250], which to some extent is suppressed [252], causing an averaging effect when filling the background. Then, the separation of background and foreground generated compelling results at low resolution [253] using a hybrid model, where the difference between frames is not significant. Thus, a pretrained model [254] using VQ-VAE resolves the previous issues. Contrastively, GAN networks generate sharp results utilizing 2D [256,257] or 3D convolutions [253,255], where the latter generates poor frames of fixed lengths. Further changes such as modifying discriminators [255,257], exploring latent paths [258], and harnessing T2I [259] show advancements in the field. Given the pioneering steps in the T2V task, these methods are supervision-based and cannot perform well for out-domain scenes and instructions.

**Quality of text features** An essential aspect of T2Vo is text. As seen from Table 2, it is evident that most studies use LSTM or GRU trained with CNN for the joint embedding between the text and visual domain. Some recent works, however, employ more powerful transformer-based models such as BERT [45], CLIP [224], and Roberta [261]. So, one future direction could be the use of the latent space of these transformer-based models for text embeddings. Another interpretation in text embeddings is the level of attention, ranging from sentences [7] to words [34], and aspects [136], where further extensions such as sensitivity to grammar, positional, and numerical information are neglected. Therefore, the new level of attention to the models is seemingly interesting.

**Power of visual models** Currently, CNN is the most-investigated model for learning visual features. Studies that relate text embedding with visual data are trained on these models. However, CNN has some significant drawbacks, including the difficulty of understanding variance in data, adversarial challenges, lack of coordinate frame, and other minor ones. To deal with these challenges, other promising networks such as [77,79,404] are presented. So, instead of CNN-based models for T2Vo, one could experiment with such models to expand the horizon of this task. Additionally, the study of knowledge distillation concept [76] for T2Vo is another promising direction, where, using the large teacher model, we can learn a better student model.

**Equality among generativity** Among various T2Vo tasks and generative models, GAN has received the most attention, whereas other undermined generative models such as autoregressive [96,263,405], VAE [89,406], flow-based [282,283], and transformer-based [39,79] models equally possess the capability for a promising future. Over the years, researchers have resolved many GAN-related issues, such as unstable training, inaccurate density estimation, lack of intrinsic metric evaluation [407], and difficulty in inversion for better understanding. However, due to the limited examination of other models, sequential learning [96], blurry results [408], fixed-length data [409], and high computational cost [410] are currently the main focus of research. By resolving these difficulties, we believe we can improve current T2Vo tasks. Notably, the evaluation metrics of IS and FID cannot be used with other models as they penalize them [411], requiring the need for a model-agnostic comparison method.

**Difficulty in perception** Among T2Vo tasks, only T2I achieves exceptional work, where the best results are for object-centric and fixed domain datasets. Consequently, this highlights the gap in the current research: there is still a struggle to understand the textual data for semantically generating complex visual output. Some studies make use of intermediate tasks, such as layout generation [43], segmentation masks [180], and the distinction between background–foreground [187], to resolve such problems. In our opinion, the use of such side information is helpful but requires additional annotation, which in the real world is very costly and limits the use of large datasets. Hence, we think that instead of densely annotating data, utilizing models that predict the added information, such as [180,184], as an intermediate task can lead to better results.

### 11.3. Cross-Modal Datasets

In dealing with deep learning models, which are data-driven, the inclusion of cross-modal datasets is necessary for deciding the future. So, in summary, we discuss the datasets with the flaws and possible solutions.

**Importance of uni-modal datasets** Studies that employ the power of pre-trained CNN models, such as VGG19 [70] trained on ImageNet [353], for visual feature extraction, are of significant use, especially for tasks that require the understanding of the visual content for either manipulation or recurrent generation such as stories and videos. Therefore, the need for massive unimodal datasets for better learning is required. Moreover, a challenging aspect of cross-modal datasets is the costly collection, where separate data are is readily available without quality annotation. So, instead of utilizing high-quality data, we can leverage the massive raw internet data of text and visual domains to train models. Afterward, these pre-trained models are easy to finetune through transfer learning for the specific task.

**Simple vs. Complex datasets** Among the variety of datasets, the use of object-centric datasets such as CUB and Oxford [322,323] proved to be valuable in evaluating different models, mostly T2I, under the less-complex scenario. Prominently, the experiments on the CUB dataset are more common than those on Oxford-102 due to the lower amount of data in Oxford, where both pose the same meaning. As a deviation from the commonly used datasets, some studies also used other similar datasets of high quality, such as Celeb-HQ and its variants, where textual annotations are currently not open-sourced. In the case of stories and videos, simple datasets such as CLEVR [337] and MNIST [412] serve as the base to evaluate the concepts, which is far from a real-world application. Comparatively, complex datasets such as MS-COCO [354], Pororo [372], and Howto100M [310] offer more challenges to current models. Practically, the limitation of low resolution, high storage requirements, and limited annotations are critical for better visual quality.

**Synthetic and real data** When dealing with T2Vo tasks, the moderately mature T2I task can handle real-world data, whereas the emerging T2S and T2V techniques are generally limited to synthetic datasets. However, some recent studies [253] utilize the massive Youtube data for T2V, where T2S still is limited to the synthetic dataset of Pororo and CLEVR. So, by collecting realistic images as a sequence of stories, T2S can be explored further.

**Dense annotations** One attribute of high-quality data is the accurate dense annotation used for supervised learning of T2Vo models. This poses two issues, the first being a quality check of the data, as [413] analyzed that human captions miss the obvious visual content, and the second is the lack of dense annotations for better learning of models, where the use of multiple captions, locally related text [171], and other costly annotations of segmentation masks, etc., are missing or not easily possible. Moreover, in the case of the joint text–visual embeddings, generally, the datasets are created by describing the contents of the visual data, which targets visual-to-text tasks and ignores the creativity and importance of the text-to-visual task. Following all this, we suggest the use of raw internet data to annotate with a deep learning model verified by human annotations from a sample of data and employ it in T2Vo tasks.

**Multiple modalities** Following humans, the need for additional modalities such as audio can allow further improvement at the cost of greater complexity. Moreover, currently, textual data are in English only, whereas the need for a multilingual model to analyze the generalizability can be one factor to consider for future research.

### 11.4. Evaluation Techniques

One of the biggest challenges in T2Vo tasks is the lack of reliable standard automated metrics for properly evaluating different tasks. At present, because of the diverse aspects of the generative models, one could optimize a metric to a specific generative model, but generalizability to all is hard. The evaluation metrics for T2Vo should serve as a guide to effectively compare the results among different models in terms of quality and semantic alignment between text and visual data.

**Quality vs. Semantics** For visual quality, ref. [414] provides a list of attributes that a metric should pose, including diversity, disentangled representation, invariance to small perturbations, closeness to human evaluation, and low complexity. In contrast to visual quality, semantic alignment between text–visual data is ambiguous due to the one-to-many mapping problems. Since the visual form is high-dimensional while natural language is in a low-dimensional convenient form, it is impossible to understand the exact meaning. Various evaluation metrics found in the literature now offer a solution to some of the listed problems, but for future preferences, these should be well defined as:Evaluate the correctness between the image and caption;Evaluate the presence of defined objects in the image;Clarify the difference between the foreground and background;Evaluate the overall consistency between previous output and successive caption;Evaluate the consistency between the frames considering the spatio-temporal dynamics inherent in videos [415].

**Improvements to current evaluations** The evaluation metrics of T2Vo are mostly IS and FID, whereas some use FSD for stories and FVD for videos. Additionally, the work on text–visual semantics utilizes the metrics of R-precision, SSIM, SOA, and captioning models. As our suggestion, we imply the use of other metrics such as LPIPS, SceneFID, and precision-recall metrics as used in some of the studies. Moreover, large pretrained models with high accuracy can be used for evaluation [416]. Separate from the automatic evaluation metrics, user studies are also frequent in analyzing the quality of the generated output but are not standardized. Thus, we suggest standardizing such techniques as in [417] for better comparison. Finally, the lack of open-source coding or clearly stating the evaluation methods puts a barrier in the way of gaining a complete understanding that could possibly explain the question of inconsistencies between different studies.

## 12. Conclusions

In this review article, we presented a broad taxonomy of text-to-visual output describing the state-of-the-art T2I, T2S, and T2V methods with follow-up modifications. We highlight the different datasets used for these methods with inefficient proposed evaluation metrics and discuss the current challenges with future directions. Our taxonomy generalizes the text-to-image synthesis task to a more comprehensive study that includes text-to-image, -story, and -video for 2D and 3D, emphasizing the research gap. Further categorization of these tasks is based on four types of emerging deep learning models—energy-based, autoregressive, GAN, and VAE—which are capable of generating novel outputs rather than retrieving an existing one, like in retrieval-based tasks. As mentioned in the paper, T2I has experienced extensive research, especially with GAN. Thus, for T2I with GAN, we build upon the previous works and complement them with the latest and more diverse studies. In short, for T2I with GAN, leveraging additional information achieves the best quality. However, other recent models such as VQ-VAE and autoregressive transformers show a promising future for T2I, addressing the limitations of GAN, including the unstable and expensive training. The other visual domains, such as story, video, or higher-dimensional output, suffer from limited research, with a more narrow focus on GAN because of the natural development of GAN from T2I. Moreover, text-guided visual manipulation is also limited to the GAN models, leaving a research gap for the future.

Lastly, we accentuate the common challenges when dealing with the T2Vo task. Following this, we brief with an in-depth discussion and future directions for the open challenges across multiple dimensions. For models, we expect the use of more diverse models other than GAN that can enhance the quality with better scene understanding. In terms of the existing datasets, our intuition is that visually grounded captions with dense crossmodal associations can improve the joint representation learning. Furthermore, we believe that through a standard and reliable evaluation metric for this domain, we can accurately define continual progress.

Significant research in the text-guided visual domain has progressed to practical implementation in various applications. Despite the progress, there is a lot of potential for improvement in terms of quality, resolution, semantics, consistency, diversity, user control, reliable automatic evaluations, standard human evaluations, and user-friendly interfaces. From this review, we aim to help researchers gain an insight into the emerging technologies for the text-guided visual domain by understanding the current SOTA methods that highlight the open challenges for future advances in the field.

## Figures and Tables

**Figure 1 sensors-22-06816-f001:**
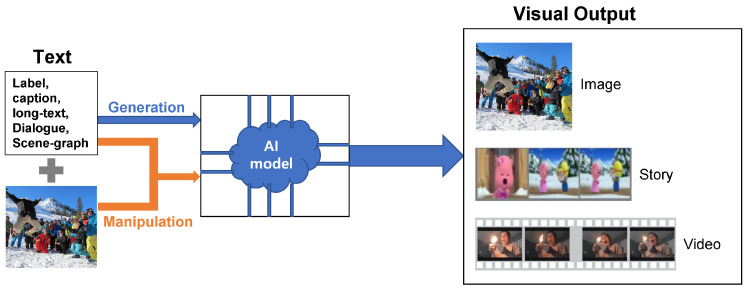
Task of the survey.

**Figure 2 sensors-22-06816-f002:**
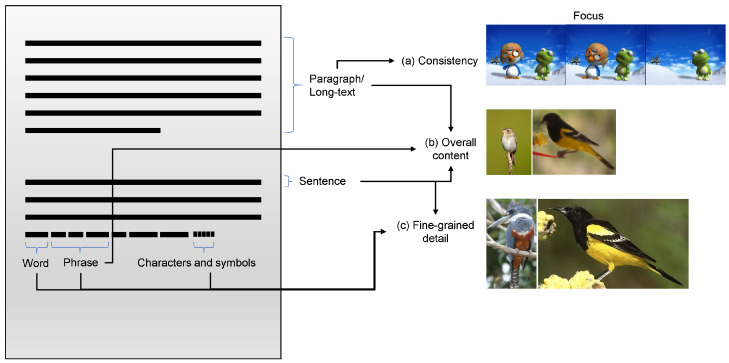
Representation of different types of text levels used in the T2Vo tasks, where different levels pay attention to one or more output detail. Figures obtained from (a) PororoQA, (b,c) CUB 2011 datasets, which represent the concept used in (a) [31], (b) [7,32], and (c) [33,34].

**Figure 3 sensors-22-06816-f003:**
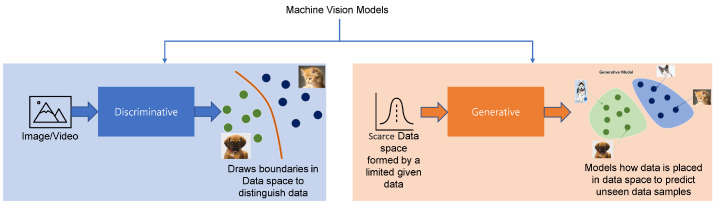
A high-level division of the deep learning vision models based on output type highlights the difference between them.

**Figure 4 sensors-22-06816-f004:**
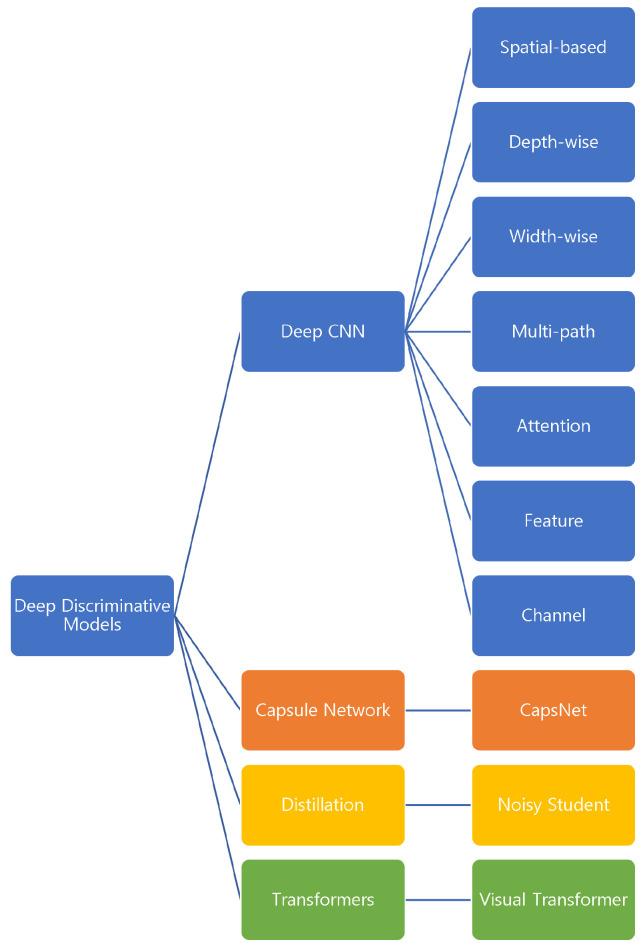
Classification of deep learning discriminative models based on their architecture, extending [80], where Deep CNN is the most studied topic for T2Vo tasks and is further categorized.

**Figure 5 sensors-22-06816-f005:**
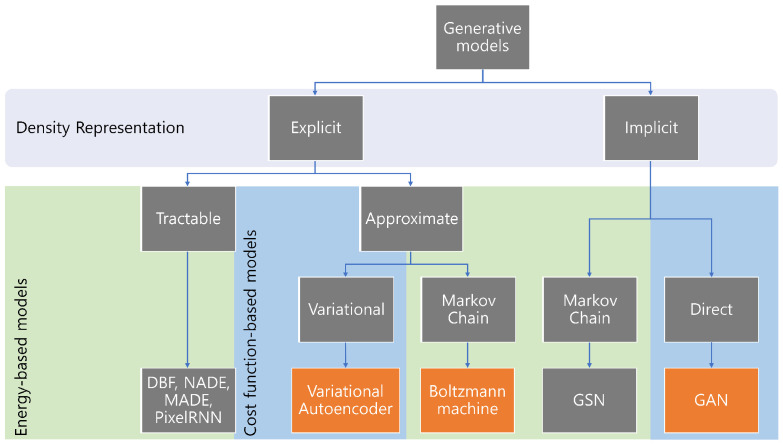
Classification of generative models: this figure is modified version of [82,83]. Orange color specifies the most popular models.

**Figure 6 sensors-22-06816-f006:**
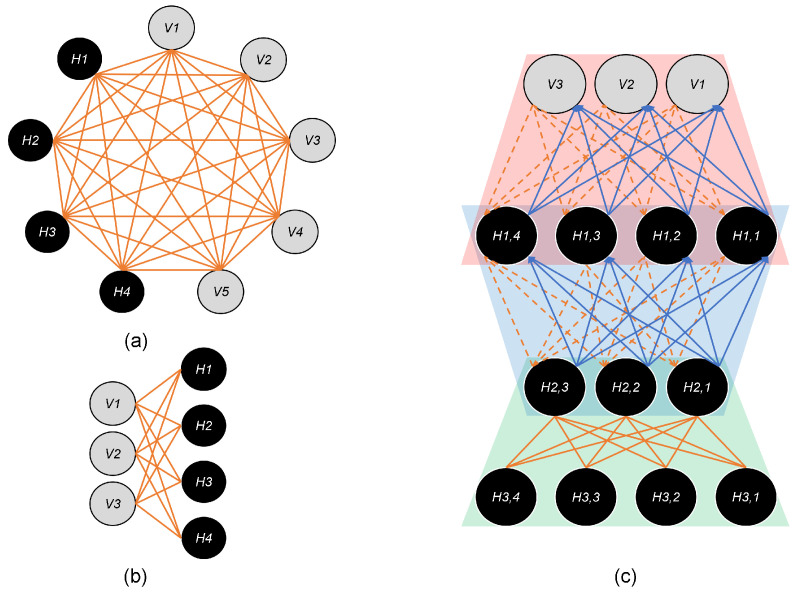
Boltzmann machine (BM) and its variants, where black and white nodes represent hidden and visible layers, respectively. (**a**) Original BM with all undirected connections. (**b**) Restricted Boltzmann Machine (RBM) with fewer connections. (**c**) Three 2-layer RBMs (Red, Blue, Green) combine to form a Deep belief Network (DBN), with the top 2 layers having directed connections (blue). However, when all connections are undirected (orange), a Deep Boltzmann machine (DBM) is formed.

**Figure 7 sensors-22-06816-f007:**
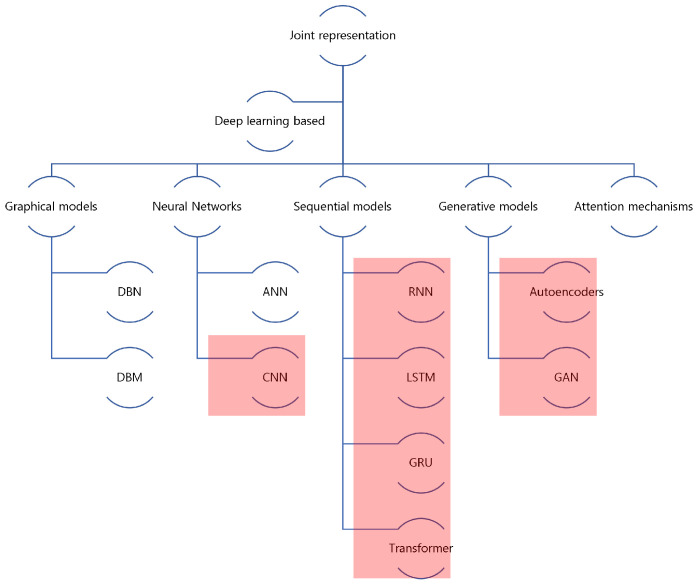
Hierarchy of joint-representation.

**Figure 8 sensors-22-06816-f008:**
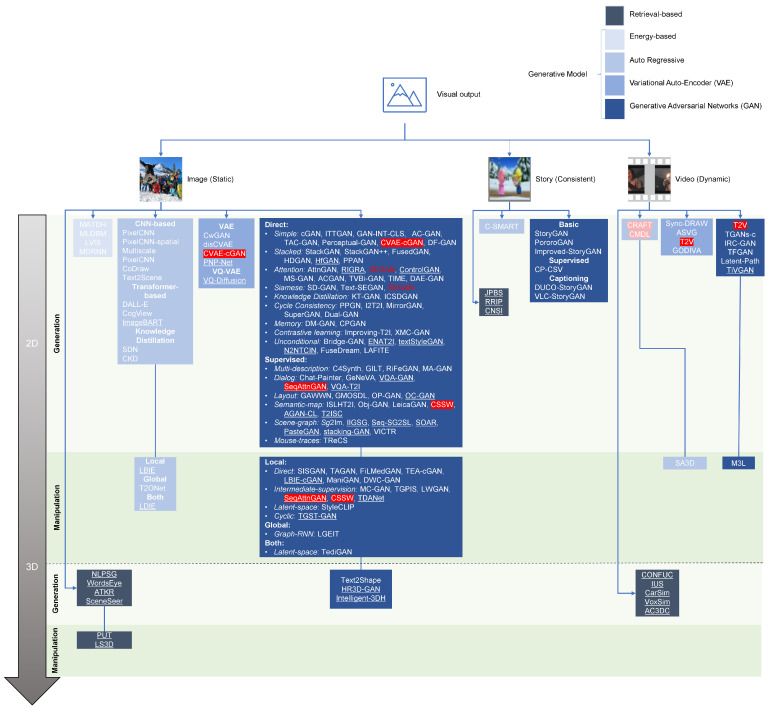
Taxonomy of Text-to-Visual output, where dark red markings indicate repetitions and light ones indicate the exceptional cases mentioned in the text. The papers listed in gray boxes are shown for the sake of completion and are not discussed in the paper.

**Figure 9 sensors-22-06816-f009:**
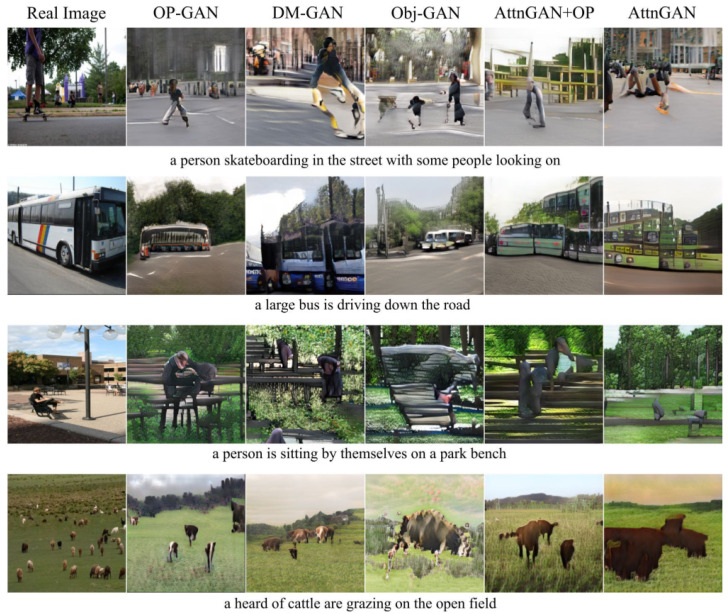
Examples reproduced from [177] using the COCO dataset.

**Figure 10 sensors-22-06816-f010:**
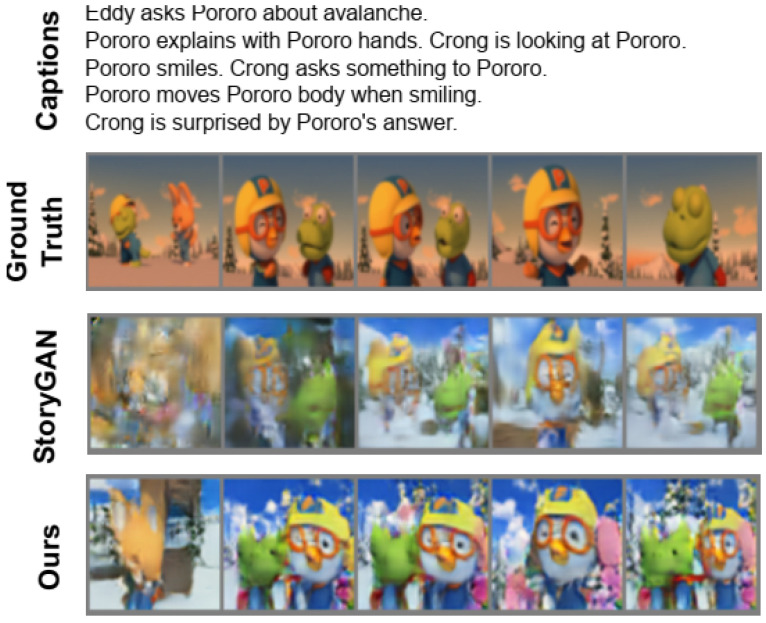
Example of Pororo-SV dataset, reproduced from [237].

**Figure 11 sensors-22-06816-f011:**
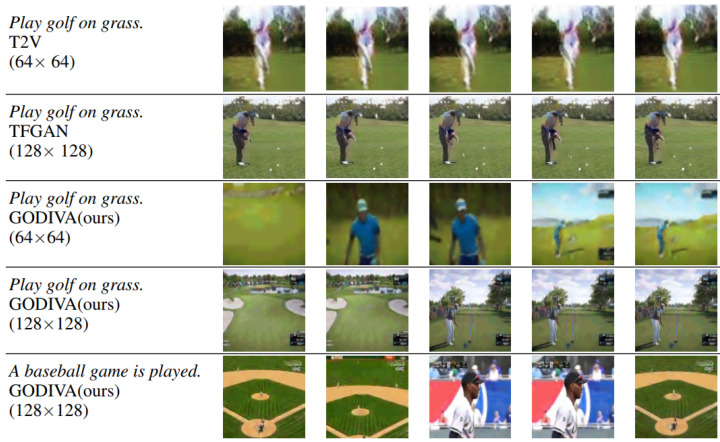
Reproduced from [254] on MSR-VTT dataset.

**Table 1 sensors-22-06816-t001:** Targeted venue for manual search.

Venue	Acronym	Selected Publications
Computer Vision and Pattern Recognition	CVPR	46
International Conference on Computer Vision	ICCV	11
Advances in Neural Information Processing Systems	NeurIPS	12
AAAI Conference on Artificial Intelligence	AAAI	5
International Conference on Machine Learning	ICML	5
European Conference on Computer Vision	ECCV	9
International Joint Conferences on Artificial Intelligence	IJCAI	2
International Conference on Learning Representations	ICLR	5

**Table 2 sensors-22-06816-t002:** Characteristics of the listed models. The references highlighted in red represent studies previously mentioned in [11], whereas those highlighted in blue are studies used for 3D visual output, as given in Figure 8.

		Model	Abbreviation	Year (Final-Version)	Characteristics	Modalities 1. Input 2. Output	Neural Networks 1. Text Model 2. Visual Model
Static (Image)	Generation	Energy-based					
		[92]	MATDH	2005	Multi-wing harmonium based on 2-layer random fields, contrastive divergence, and variational algorithm for learning	label, text image	Poisson model + bag-of-words Color-histogram by Guassian model
		[93]	MLDBM	2012	Bi-modal DBM as a generative model for learning joint representation of multimodal data	label, text image	text-DBM image-DBM
		[94]	LVIS	2013	Conditional Random Field for generating scenes by handling 2 nouns and 1 relation, Abstract Scene dataset through MTurk	caption image	Sentence parsing Conditional Random-Field
		[95]	MDRNN	2014	RBM with contrastive divergence and multi-prediction training, Minimize variation of information, Recurrent encoding structure for finetuning	label image	text-RBM image-RBM
		Autoregressive					
		[96]	PixelCNN	2016	Multi-conditioning (label, latent embeddings), work as Image autoencoder, gated-CNN layer	label image	one-hot encoding Gated-CNN
		[97]	PixelCNN-spatial	2017	PixelCNN for T2I with controllable object location using segmentation masks or keypoints	caption+segmentation-map/ keypoint-map image	char-CNN-GRU CNN (for seg/keypoint), modified PixelCNN
		[98]	Multiscale PixelCNN	2017	Parallel PixelCNN through conditionally independent pixel groups, multiscale image generation, multiple tasks (label-image, T2I, action-video)	caption + keypoint-map, label, action-video image: 512 × 512, video-frame: GRP	char-CNN-GRU ResNet+PixelCNN, Conv-LSTM (for video)
		[99]	CoDraw	2019	Collaborative image drawing game for CoDraw dataset, Multiple AI models for depicting Tell-Draw game	caption+previous-image abstract scene	LSTM-attention BiLSTM, Reinforcement learning
		[100]	Text2Scene	2019	Seq2seq framework, ConvGRU for recurrent drawing, 2 attention-based decoders, unified framework for 3 generation tasks (cartoon, semantic layout, image)	caption+ previous-image=>layout image: 256 × 256	BiGRU CNN+ConvGRU (for previous)
		[101]	DALL-E	2021	Large pre-trained Autoregressive transformer, zero-shot learning, generates 512 images/caption and selects 1 through CLIP	caption image: 256 × 256	256-BPE+CLIP discrete-VAE+ResNet+Transformer [102]
		[103]	CogView	2021	Large pre-trained GPT-transformer with VQ-VAE, caption-loss evaluation metric, PB-relaxation and Sandwich-LN to stablize training, zero-shot generation, self-reranking to avoid CLIP	captions image: 256 × 256	{SentencePiece [104] VQ-VAE}=>GPT
		[105]	ImageBART	2021	The hierarchical bidirectional contextualized into autoregressive transformer model, inverting multinomial diffusion by Markov chain, addressing unidirectional and single-scale limitations	caption, label, previous-image image: 256 × 256, 300 × 1800	CLIP CNN+Markov-Chain + Transformer [39]
		Knowledge Distillation					
		[106]	SDN	2018	T2I using a Distillation network with VGG19 as a teacher and a similar student generative model, 2-stage training with different distillation	caption+real-image image: 224 × 224	char-CNN-GRU VGG19
		[107]	CKD	2019	Transfer knowledge from image classifier and captioning model, a multi-stage distillation paradigm to adapt to multiple source models	caption+real-image=>caption image: 299 × 299	Captioning model [108], Text-encoder [43] Inception-v3, VGG19
		Variational Auto-encoder					
		[5]	CwGAN	2016	Conditional alignDRAW model with soft attention mechanism, post-processing by LAPGAN	caption image: 32 × 32	BiLSTM-attention LSTM-VAE
		[109]	disCVAE	2016	Attribute to Image model, General energy minimization algorithm for posterior inference, separate image foreground, and background with layered VAE	visual attributes image: 64 × 64	multi-dimension vectors VAE
		[110]	CVAE-cGAN	2018	Context-aware stacked cross-model (CVAE, cGAN) framework, CVAE decouples foreground and background while cGAN refines it	caption image:256 × 256	char-CNN-GRU + CA CVAE-cGAN
		[111]	PNP-Net	2018	PNP-Net, a generic canonical VAE T2I framework, with zero-shot learning neural modules for modifying visual appearance of objects	tree-structure description image: 128 × 128	NMN + LSTM [112] VAE
		[113]	VQ-Diffusion	2022	Non-autoregressive Vector-quantized diffusion, VQ-VAE with denoising diffusion model, eliminates unidirectional bias and adds mask-replace diffusion to remove error accumulation	caption, label image: 256 × 256	BPE-encoding [114] + CLIP (ViT-B) VQ-VAE + Diffusion-transformer [115]
		GAN-based					
		Direct					
		Simple					
		[4]	Conditional GAN	2014	Introduction of conditional GAN, unimodal task of class2image, and multimodal task of image-tagging	label, tag image: 28 × 28	Skip-gram [25], one-hot-vector deep-CNN
		[116]	ITTGAN	2016	Solutions to unstable training of GAN, focuses on 2 applications of GAN, introduces IS evaluation metric and human evaluation through MTurk	label image: 128 × 128	one-hot encoder cGAN
		[7]	GAN-INT-CLS	2016	Introduction to GAN model for T2I, matching-aware discriminator, manifold interpolation regularizer for text in generator, showed style-transfer	caption image: 64 × 64	char-CNN-LSTM DC-GAN
		[117]	AC-GAN	2017	Improved training of T2I GAN, label-prediction discriminator, higher-resolution, introduces MS-SSIM evaluation metric, identify GAN issues	label image: 128 × 128, 64 × 64; class-label	one-hot vector AC-GAN
		[118]	TAC-GAN	2017	Improving perceptual quality, the generator optimizes contextual and perceptual loss	caption image: 64 × 64	char-CNN-RNN DC-GAN
		[119]	Perceptual-GAN	2017	AC-GAN for T2I	caption, label image: 128 × 128	Skip-thought DCGAN
		[120]	Text2Shape	2018	End-to-End framework for text-to-3D Shape, joint representation for retrieval, generation with conditional wassertein GAN, 2 datasets as Primitives and ShapeNetCore	description voxels: 32 × 32 × 32	CNN-RNN (GRU) 3D-CNN + Wassertian-GAN
		[121]	HR3D-GAN	2019	2-stage high-resolution GAN model for voxels, critic-net for multiple roles, multiple indices for comparison	description voxels: 64 × 64 × 64	Text2Shape
		[122]	Intelligent-3DH	2020	House-plan-generative model (HPGM), new dataset as Text-to-3D house model, 2-subtasks (floor-plan by GC-LPN, interior textures by LCT-GAN),	description=>scene-graph texture-images: 160 × 160; floor-plan: drawings	Scene-graph-parser [123] Graph-conv-net + Bounding box-regression + text-image-GAN
		[124]	DF-GAN	2021	1-stage T2I for high-resolution, text-image fusion block, skip-z with truncation, target-aware discriminator having matching-aware–gradient-penalty (MA-GP)	caption image: 256 × 256	Bi-LSTM-Inception-v3 (DAMSM) + CA unconditional GAN (Geometric-GAN)
		Stacked					
		[32]	StackGAN	2017	Stacked-GAN for high-resolution T2I, introduces conditioning augmentation for text, improved details, and diversity	caption image: 256 × 256	char-CNN-RNN (pre-train) + CA residual-CNN
		[33]	StackGAN++	2018	Multi-stage tree-like GAN design, the t-SNE algorithm used for identifying mode-collapse, stability by multiscale image distribution, and conditional–unconditional joint distribution	caption image: 256 × 256	char-CNN-RNN (pre-train) + CA residual-CNN
		[125]	FusedGAN	2018	2-fused generators (conditional, unconditional), sampling images with controlled diversity, semi-supervised data for training, avoids additional intermediate information	caption image: 64 × 64	char-CNN-RNN + CA DC-GAN
		[126]	HDGAN	2018	Depth-wise adversarial learning using one hierarchical generator and multiple discriminators, higher resolution, multi-purpose adversarial loss, introduces the VSSM evaluation metric	caption image: 512 × 512	char-CNN-RNN (pre-train) + CA res-CNN
		[127]	PPAN	2019	1-pyramid generator with 3 discriminators for feed-forward coarse-to-fine generation, perceptual loss for semantic similarity, multi-purpose discriminator for consistency and invariance	caption image: 256 × 256	char-RNN (pre-train) + CA residual-CNN
		[128]	HfGAN	2019	Hierarchically fused GAN with 1 discriminator, generator fuses features based on residual learning, local–global feature separation, skip connection to avoid degradation problem	caption image: 256 × 256	DAMSM + CA DC-GAN
		Attention					
		[34]	AttnGAN	2017	Attention model for T2I with multi-stage attentional-generative-network (AGN), deep-attentional-multimodal-similarity-model (DAMSM) for image–text matching loss	caption image: 256 × 256	DAMSM + CA Attn-GAN
		[129]	RIGRA	2019	Shows regular-grid region with word-attention causes problems, introduces true-grid attention regions by auxiliary bounding boxes and phrases, considered word phrases	caption image: 256 × 256	DAMSM + phrase-LSTM + CA Attn-GAN
		[130]	SEGAN	2019	Semantic-consistency module (SCM) for image consistency, sliding loss to replace contrastive loss, Attention-competition module (ACM) for adaptive attention weights, Siamese net with 2 semantic similarities	caption image: 256 × 256	DAMSM + Cross-modal-similarity (ACM) + CA Attn-GAN
		[131]	Control-GAN	2019	High quality, controlled part generation, word-level spatial and channel-wise attention-driven generator, word-level discriminator, adoption of perceptual loss	caption image: 256 × 256	DAMSM + CA AttnGAN
		[132]	MS-GAN	2019	Multi-stage attention-modulated generators (AMG), similarity-aware discriminators (SAD)	caption image: 256 × 256	DAMSM + CA 3-stage GAN
		[133]	ACGAN	2020	Attentional concatenation with multilevel cascaded structure, higher resolution, minibatch discrimination for discriminator to increase diversity	caption image: 1024 × 1024	DAMSM + CA Residual-CNN
		[134]	TVBi-GAN	2020	Consistency by 2 semantic-enhanced modules, Semantic-enhanced attention (SEAttn) for realism, Semantic-enhanced batch normalization (SEBN) to balance consistency and diversity	caption image: 256 × 256	DAMSM + CA Deep-CNN
		[135]	TIME	2020	Avoiding pre-trained models by end-to-end Transformer training, and sentence-level text features are 2 unnecessary techniques (2D positional, hinge loss) for better attention and learning paces	caption image: 256 × 256	Transformer Transformer-modified + AttnGAN
		[136]	DAE-GAN	2021	Multiple granuality text representation (sentence, word, aspect), Aspect-aware Dynamic Re-drawer (ADR), ADR from Attended Global Refinement (AGR), and Aspect-aware Local Refinement (ALR)	caption image: 256 × 256	(LSTM+DAMSM+CA) + NLTK (POS-tagging) Inception_v3
		Siamese					
		[137]	Text-SEGAN	2019	Focused text semantics by 2 components, Siamese mechanism in discriminator for high-level semantics, semantic-conditioned-batch-normalization for low-level semantics	caption image: 256 × 256	Bi-LSTM Semantic-Conditioned Batch Normalization (SCBN)
		[138]	SD-GAN	2019	Avoids mode-collapse, AC-GAN discriminator measuring semantic relevance instead of class prediction, training triplet with positive–negative sampling to improve training	caption image: 64 × 64	char-CNN-RNN GAN-INT-CLS
		Knowledge Distillation					
		[139]	KT-GAN	2020	Semantic distillation mechanism (SDM) for teaching text-encoder in T2I through image-encoder in I2I, Attention-transfer mechanism updates word and subregions attention weights	caption image: 256 × 256	BiLSTM+DAMSM+CA AttnGAN + AATM + SDM-(I2I+T2I)
		[140]	ICSDGAN	2021	Interstage knowledge distillation, cross-sample similarity distillation (CSD) blocks	caption image: 256 × 256	Bi-LSTM [34] MS-GAN [132]
		Cycle Consistency					
		[141]	PPGN	2017	Prior on latent improves quality and diversity, unified probabilistic interpretation of related methods, shows multi-condition generation, improves inpainting, modality-agnostic approach	caption, label, latent image: 227 × 227	2-layer LSTM AlexNet DNN, MFV
		[142]	I2T2I	2017	Novel training method by T2I-I2T for T2I, 3-module network (image-captioning, image-text mapping, GAN), textual data augmentation by image-captioning module	caption image: 64 × 64	LSTM Inception_v3 + GAN-CLS
		[143]	MirrorGAN	2019	Semantic text-embedding module (STEM), global–local attentive cascaded module (GLAM), semantic text regeneration and alignment module (STREAM), Cross-entropy-based loss	caption image: 256 × 256; caption	DAMSM+CA Attn-GAN, CNN-RNN
		[144]	SuperGAN	2019	Adoption of the cycle-GAN framework, 2 main components (synthesis and captioning), cycle-consistent adversarial loss and training strategy, new color-histogram evaluation metric	caption image: 128 × 128; caption	Skip-thought StackGAN, AlexNet-LSTM
		[145]	Dual-GAN	2019	Introduction of latent space disentangling of content and style, dual inference mechanism, content learned in a supervised and unsupervised way, style only unsupervised	caption=>latent-space image: 64 × 64	char-CNN-RNN + CA HDGAN, BiGAN
		Memory					
		[146]	DM-GAN	2019	Dynamic memory-based model for high-quality images when initial image is fuzzy, memory writing gate for selecting relevant word, response gate to fuse image–memory information	caption image: 256 × 256	parameter-fix-DAMSM + CA KV-MemNN [147] + GAN
		[148]	CPGAN	2020	Memory structure to parse textual content during encoding, Memory-Attented Text encoder, Object-aware Image encoder, Fine-grained conditional discriminator	memory+caption image: 256 × 256	Bi-LSTM + DAMSM + (Yolo_v3+BUTD)=>memory Yolo-v3 + AttnGAN [149]
		Contrastive learning					
		[150]	Improving-T2I	2021	Contrastive learning for semantically consistent visual–textual representation, synthetic image consistency in GAN, flexible to be fitted in previous methods	caption image: 256 × 256	BiLSTM + DAMSM + contrastive-learning Inception_v3 + (AttnGAN, DMGAN)
		[151]	XMC-GAN	2022	Single-stage GAN with several contrastive losses, benchmark on OpenImages dataset	caption image: 256 × 256	BERT conditional-GAN + VGG
		Unconditional					
		[152]	Bridge-GAN	2019	Transitional space as a bridge for content-consistency, 2 subnetworks (Transitional mapping and GAN), ternary mutual information objective function for optimizing transitional space	caption image: 256 × 256	char-CNN-RNN Transitional-mapping + GAN
		[153]	ENAT2I	2020	Single-stage architecture with 1 G/D using residual net, text image editing via arithmetic operations, sentence interpolation technique for smooth conditional space, and augmentation	caption image: 256 × 256	modified-DAMSM (BiGRU with global-vector only) + Sentence-Interpolation (SI) Bi-GAN-deep [154]
		[155]	textStyleGAN	2020	Unifying pipeline (generation manipulation), a new dataset of CelebTD-HQ with faces and descriptions, pre-trained weight manipulation of textStyleGAN for facial image manipulation	caption, attribute image (T2I and A2I): 256 × 256, 1024 × 1024	pre-train a Bi-LSTM-CNN-CMPM-CMPC [156] + CA StyleGAN [157]
		[158]	N2NTCIN	2020	Reuse of expert model for multimodality, a flexible conditionally invertible-domain-translation-network (cINN), computationally affordable synthesis, and generic domain transfer	caption, attribute image (T2I and A2I): 256 × 256	BERT BigGAN
		[159]	FuseDream	2021	CLIP+GAN space, zero-shot learning, 3-techniques to improve (AugCLIP score, initialization strategy, bi-level optimization)	caption image: 512 × 512	CLIP + AugCLIP BiGAN
		[160]	LAFITE	2022	T2I in various settings (Language-free, zero-shot, and supervised), VinVL [161] as image captioning for T2I, reduced model size	caption image: 256 × 256	CLIP StyleGAN2 + ViT
		Supervised					
		Multiple descriptions					
		[162]	C4Synth	2018	Introduced multi-caption T2I, 2 models as C4Synth and Recrrenct-C4Synth, Recurrent model removes caption limitation, also tested for image style transfer	multi-captions image: 256 × 256	char-CNN-RNN + CA CycleGAN + RecurrentGAN
		[163]	GILT	2019	Introduced indirect long-text T2I, comparing 2 embedding types (no-regularize and regularize), NOREG for image generation, REG for classification	sentences (instructions+ingredients) image: 256 × 256	ACME [164] StackGAN-v2
		[165]	RiFeGAN	2020	Attention-based caption-matching model to avoid conflicts and enrich from prior knowledge, self-attentional embedding mixtures (SAEM) for features from enriching captions, high quality	multi-captions image: 256 × 256	RE2 [166] + BiLSTM + CA + SAEM + MultiCap-DAMSM AttnGAN
		[167]	MA-GAN	2021	Captures semantic correlation between sentences, progressive negative sample selection mechanism (PNSS), single-sentence generation and multi-sentence discriminator module (SGMD)	multi-senteces image: 256 × 256	AttnGAN + CA AttnGAN
		Dialog					
		[168]	Chat-Painter	2018	High quality using VisDial dialogues and MS-COCO captions, highlights GAN problems (object-centric, mode-collapse, unstable, no end-to-end training)	caption+dialogue image: 256 × 256	char-CNN-RNN (caption), Skip-Thought-BiLSTM (dialogue) StackGAN
		[169]	GeNeVA	2019	Recurrent-GAN architecture~Generative Neural Visual Artist (GeNeVA), new i-CLEVR dataset, new relationship similarity evaluation metric	sequential-text+prev-image image: computer-graphics	GloVe [170] + BiGRU shallow-residual-CNN
		[171]	VQA-GAN	2020	Introduced QA with locally related text for T2I, new Visual-QA accuracy evaluation metric, 3-module model (heirarchical QA encoder, QA-conditional GAN, external VQA loss)	Visual-QA+layout+label image: 128 × 128	2-level-BiLSTM + CA + DAMSM AttnGAN-EVQA (Global-local pathway)
		[172]	SeqAttnGAN	2020	Introduced interactive image editing with sequential multi-turn textual commands, Neural state tracker for previous images and text, 2 new datasets such as Zap-Seq and DeepFashion-Seq	image, sequential-interaction image: 64 × 64	Bi-LSTM + RNN-GRU + DAMSM modified-AttnGAN (multi-scale joint G-D)
		[173]	VQA-T2I	2020	Combining AttnGAN with VQA [174] to improve quality and image–text alignment, utilizing VQA 2.0 dataset, create additional training samples by concatenating QA pairs	caption+QA image: 256 × 256	Bi-LSTM + DAMSM AttnGAN + VQA [174]
		Layout					
		[43]	GAWWN	2016	Text-location control T2I for high-resolution, text-conditional object part completion model, new dataset for pose-conditional text–human image synthesis	caption+Bounding box/ keypoint image: 128 × 128	char-CNN-GRU (average of 4 captions) Global-local pathway
		[175]	GMOSDL	2019	Fine-grained layout control by iterative object pathway in generator and discriminator, only bounding box and label used for generation, added discriminator for semantic location	(caption+label)=>layout image: 256 × 256	char-CNN-RNN + one-hot vector + CA (StackGAN+AttnGAN) + STN [176]
		[177]	OP-GAN	2020	Model having object-global pathways for complex scenes, new evaluation metric called Semantic object accuracy (SOA) based on pre-trained object detector	(caption+label)=>layout image: 256 × 256	RNN-encoder + DAMSM AttnGAN
		[178]	OC-GAN	2020	Scene-graph similarity module (SGSM) improves layout fidelity, mitigates spurious objects and merged objects, conditioning instance boundaries generates sharp objects, new SceneFID evaluation metric	scene-graph+boundry-map +layout image: 256 × 256	GCN [179] + Inception-v3 SGSM
		Semantic-map					
		[180]	ISLHT2I	2018	Heirarchical approach for T2I inferring semantic layout, improves image–text semantics, sequential 3-step image generation (bbox-layout-image)	caption=>(label+bbox)=>mask image: 128 × 128	char-CNN-RNN LSTM with GMM [181], Bi-convLSTM [182], Generative-model [183]
		[184]	Obj-GAN	2019	Object-centered T2I with layout-image generation, object-driven attentive generator, new fast R-CNN-based object-wise discriminator, improved complex scenes	caption=>(label+bbox) =>shape image: 256 × 256	Bi-LSTM + DAMSM + GloVe attentive-seq2seq [185], Bi-convLSTM, 2-Stage GAN
		[186]	LeicaGAN	2019	Textual–visual co-embedding network (TVE) containing text–image and text–mask encoder, mulitple prior aggregation net (MPA), cascaded attentative generator (CAG) for local–global features	caption=>mask image: 299 × 299	Bi-LSTM Inception-v3
		[187]	CSSW	2020	Introduced weakly supervised approach, 3 inputs (maps, text, labels), foreground–background generation, resolution-independent attention module, semantic-map to label maps by the object detector	caption+attributes +semantic-map image: 256 × 256	BERT, bag-of-embeddings (class+attribute) SPADE [188]
		[189]	AGAN-CL	2020	Model to improve realism, the generator has 2 sub-nets (contextual net for generating contours, cycle transformation autoencoder for contour-to-images), injection of contour in image generation	caption=>contour image: 128 × 128	CNN-RNN VGG16, Cycle-transformation-autoencoder ([190] + ResNet)
		[191]	T2ISC	2020	End-to-End T2I framework with spatial constraints targetting multiple objects, synthesis module taking semantic and spatial information to generate an image	caption=>layout image: 256 × 256	BiLSTM Multi-stage GAN
		Scene-graph					
		[192]	Sg2Im	2018	Introduced Scene-graph-to-image, graph-convolution net for processing input, generates layout by Bounding box and segmentation mask, cascaded-refinement net for layout-to-image	scene-graph=>layout image: 64 × 64, 128 × 128	Scene-graph [123] Graph-convolution-Net (GCN) + Layout-prediction-Net (LPN) + Cascaded-refinement-net(CRN) [193]
		[194]	IIGSG	2019	Interactive image generation from incrementally growing scene-graph, recurrent architecture for Sg2Im generation, no–intermediate supervision required	expanding-scene-graph =>layout image: 64 × 64	Scene-graph [195] Recurrent (GCN + LPN + CRN)
		[196]	Seq-SG2SL	2019	Transformer-based model to transduce scene-graph and layout, Scene-graph for semantic-fragments, brick-action code segments (BACS) for semantic-layout, new SLEU evaluation metric	scene-graph => SF semantic-layout	Scene-graph [197] (SF+BACS => layout ) + Transformer
		[198]	SOAR	2019	Dual embedding (layout–appearance) for complex scene-graphs, diverse images controllable by user, 2 control modes per object, new architecture and loss-terms	scene-graph=>mask=>layout image: 256 × 256	Scene-graph [195] Autoencoder
		[199]	PasteGAN	2019	Object-level image manipulation through scene-graph and image-crop as input, Crop-Reining-Net and Object–Image Fuser for object interactions, crop-selector for compatible crops	scene-graph=>object-crops image: 64 × 64	Scene-graph [195] GCN + Crop-selector + crop-refining-net + object-image-fuser + CRN
		[200]	stacking-GAN	2020	Visual-relation layout module using 2 methods (comprehensive and individual), 3-pyramid GAN conditioned on layout, subject–predicate–object relation for localizing Bounding boxes	scene-graph=>layout image: 256 × 256	Scene-graph [195] GCN + comprehensive-usage-subnet + RefinedBB2Layout + conv-LSTM + GAN (CRN)
		[201]	VICTR	2020	Example of text-to-scene Graph, new visual–text representation information for T2I, text representation also for T2Vision multimodal task	caption=>scene-graph image: 256 × 256	Parser [202] + GCN AttnGAN, StackGAN, DMGAN
		Mouse-traces					
		[203]	TReCS	2021	Sequential model using grounding (mouse-traces), segmentation image generator for the final image, descriptions retrieve segmentation masks and predict labels aligned with grounding	mouse-traces+segmentation- mask + narratives image: 256 × 256	BERT Inception-v3
Static (Image)	Manipulation	Autoregressive					
		[204]	LBIE	2018	Generic framework for text-image editing (segmentation and colorization), recurrent attentive models, region-based termination gate for fusion process, new CoSaL dataset	image+description image: 512 × 512, 256 × 256	Bi-LSTM (GRU-cells) VGG, CNN
		[205]	LDIE	2020	Language-request (vague, detailed) image editing task for local and global, new GIER dataset, baseline algorithm with CNN-RNN-MAttNet	image+description image: 128 × 128 (training), variable	Bi-LSTM ResNet18 + MattNet [206]
		[207]	T2ONet	2021	Model for interpretable global editing operations, operation planning algorithm for operations and sequence, new MA5k-Req dataset, the relation of pixel supervision and Reinforcement Learning (RL)	image+description image: 128 × 128 (training), variable	GloVe + BiLSTM ResNet18
		GAN-based					
		Local					
		Direct					
		[208]	SISGAN	2017	Image manipulation using GAN (realistic, text-only changes), end–end architecture with adversarial learning, a training strategy for GAN learning	image+caption image: 64 × 64	OxfordNet-LSTM [209] + CA VGG
		[210]	TAGAN	2018	Text-adaptive discriminator for word-level local discriminators of text-attributes	image+caption image: 128 × 128	training BiGRU + CA + fastText [30] SISGAN
		[211]	FiLMedGAN	2018	cGAN model (FiLMedGAN) using Feature-wise Linear Modulation (FiLM [212]), feature transformations and skip-connections with regularization	image+caption image: 128 × 64	fastText + GRU VGG-16, SISGAN
		[213]	TEA-cGAN	2019	Two-sided attentive cGAN architecture with fine-grained attention on G/D, 2-scale generator, high resolution, Attention-fusion module	image+caption image: 256 × 256	BiLSTM + fastText AttnGAN
		[214]	LBIE-cGAN	2019	Language-based image editing (LBIE) with cGAN, conditional Bilinear Residual Layer (BRL), highlights representation learning issue for 2-order correlation between 2 conditioning vectors in cGAN	caption+caption image: 64 × 64	OxfordNet-LSTM [209] VGG, SISGAN
		[215]	ManiGAN	2020	2 key modules (ACM and DCM), ACM correlates text-relevant image regions, DCM rectifies mismatch attributes and completes missing ones, new manipulative-precision evaluation metric	image+caption image: 256 × 256	RNN (TAGAN, AttnGAN) Inception-v3 + ControlGAN
		[216]	DWC-GAN	2020	Textual command for manipulation, 3 advantages of commands (flexible, automatic, avoid need-to-know-all), disentangle content and attribute, new command annotation for CelebA and CUB	text-command+image image: 128 × 128	LSTM + Skip-gram-fastText GMM-UNIT [217]
		Intermediate supervision					
		[218]	MC-GAN	2018	Image manipulation as foreground–background by generating a new object, introduces synthesis block	image+caption+mask image: 128 × 128	char-CNN-RNN + CA StackGAN
		[219]	TGPIS	2019	Text-guided GAN-based pose inference net, new VQA-perceptual-score evaluation metric, 2-stage framework (pose-to-image) using attention-upsampling and multi-modal loss	image+pose+caption image: 256 × 256	BiLSTM Pose-encoder [220], CNN
		[221]	LWGAN	2020	Word-level discriminator for image manipulation, word-level supervisory labels, lightweight model with few parameters	image+caption image: 256 × 256	BiLSTM + CA + ACM + attention (spatial-channel) + PoS-tagging Inception-v3 + VGG-16
		[222]	TDANet	2021	Text-guided dual attention model for image inpainting, inpainting scheme for different text to obtain pularistic outputs	corrupt-image+caption image: 256 × 256	GRU (AttnGAN) ResNet
		Latent space					
		[223]	StyleCLIP	2021	3-techinques for CLIP+StyleGAN (text-guided latent-optimizer, latent-residual-mapper, global-mapper)	image+caption/attribute image: 256 × 256	CLIP + prompt-engineering [224] StyleGAN
		Cyclic					
		[225]	TGST-GAN	2021	Style transfer-based manipulation from 3 components (captioning, style generation, style-transfer net), module-based generative model	image+caption=>caption =>style-image image: -	LSTM + AttnGAN ResNet101 + AttnGAN + VGG19
		Global					
		[226]	LGEIT	2018	Global image editing with text, 3 different models (hand-crafted bucket-based, pure ended-end, filter-bank), Graph-RNN for T2I, a new dataset	image+caption image: -	GloVe + BiGRU, Graph-GRU [227] cGAN, GAN-INT-CLS, StyleBank [228]
		Both					
		[229]	TediGAN	2021	A unified framework for generation and manipulation, a new Multi-modal Celeb-HQ dataset, GAN-inversion for multi-modalities (text, sketch, segmentation-map)	caption, sketch, segmentation-mask, image image: 1024 × 1024	Text-encoder (RNN) + Visual-linguistic-similarity StyleGAN
Consistent (Stories)	Generation	Autoregressive					
		[230]	C-SMART	2022	Introduced a Bidirectional generative model using multi-modal self-attention on long-text and image as input, cyclically generated pseudo-text for training (text–image–text), high resolution	story (sequence-of-sentences) +image image-sequences: 128 × 128	Transformer VQ-VAE + Recurrent-transformer (with gated memory)
		GAN-based					
		Basic					
		[231]	StoryGAN	2019	Sequential-GAN consists of 3 components (story-encoder, RNN-based context encoder, GAN), Text2Gist module, 2 new datasets (Pororo-SV and CLEVR-SV)	story image-sequences: 64 × 64	(USE [232])-story_level + (MLP + CA + GRU + Text2Gist)-sentence_level RNN-(Text2Gist) + Seq-GAN (2-discriminators as story and image)
		[31]	PororoGAN	2019	Aligned sentence encoder (ASE) and attentional word encoder (AWE), image patches discriminator	story image-sequences: 64 × 64	StoryGAN
		[233]	Improved-StoryGAN	2020	Weighted activation degree (WAD) in discriminator for local–global consistency, dilated convolution for the limited receptive field, gated convolution for initial story encoding with BiGRU	story image-sequences: 64 × 64	USE-Gated_convolution (story-level) + BiGRU-Text2Gist (sentence-level) Dilated-convolution [234]
		Supervised					
		[235]	CP-CSV	2020	Character preserving framework for StoryGAN, 2 text-encoders for sentence and story-level input, 3 discriminators (story, image, figure segmentation), new FSD evaluation metric	story=>segmentation-maps image-sequences: 64 × 64	StoryGAN + Object-detection-model [236]
		Captioning					
		[237]	DUCO-StoryGAN	2021	Dual learning via video redescription for semantic alignment, copy transform for a consistent story, memory augmented recurrent transformer, Evaluation metrics (R-precision, BLEU, F1-score)	story image-sequences: 64 × 64	CA + (MART [238] + GRU)~context-encoder 2-stage GAN + copy-transform
		[239]	VLC-StoryGAN	2021	Model using text with commonsense, dense-captioning for training, intra-story contrastive loss between image regions and words, new FlintstonesSV dataset	story image-sequences: 64 × 64	GloVe + (MARTT+CA) + (ConceptNet [240] + Transformer-graph [241]) 2-stage GAN + Video-captioning [242]
Dynamic (Video)	Generation	Autoregressive (Retrival, dual-learning)					
		[243]	CRAFT	2018	Sequential training of Composition-Retrival-and-Fusion net (CRAFT), 3-part model (layout composer, entity retriever, background retriever), new dataset of FlintStones	caption=>layout =>entity-background retrival video: 128 × 128; frames: 8	BiLSTM CNN, MLP
		[244]	CMDL	2019	End-to-End crossmodal dual learning, dual mapping structure for bidirectional relation as text–video–text, multi-scale text-visual feature encoder for global and local representations	description=>video=> decription video: -	LSTM, (GloVe + BiLSTM [245]) 3D-CNN [246] + VGG19
		[247]	SA3D	2020	2-stage pipeline for static and animated 3D scenes from text, new IScene dataset, new multi-head decoder to extract multi-object features	description=>Layout video: computer-graphics	TransformerXL [248] (LSTM + Attn-BLock) + Blender [249]
		Variational Auto-Encoder					
		[250]	Sync-DRAW	2017	Introduced T2V task by attentive recurrent model, 3 components (read-mechanism, R-VAE, write-mechanism), a new dataset of Bouncing MNIST video with captions, and KTH with captions	caption, prev-frame video: 64 × 64, 120 × 120; frames: 10, 32	Skip-thought [251] LSTM+VAE
		[252]	ASVG	2017	Text–video generation from long-term and short-term video contexts, selectively combining information with attention	caption, prev-frame video: 64 × 64, 120 × 120; frames: 10, 15	BiLSTM-attention ConvLSTM+VAE
		[253]	T2V	2017	Hybrid text–video generation framework with CVAE and GAN, a new dataset from Youtube, intermediate gist generation helps static background, Text2Filter for dynamic motion information	caption video: 64 × 64; frames: 32	Skip-thought CVAE+GAN
		[254]	GODIVA	2021	Large text–video pretrained model with 3-dimensional sparse attention mechanism, new Relative matching evaluation metric, zero-shot learning, auto-regressive prediction	caption video: 64 × 64, 128 × 128; frames: 10	positional-text-embeddings VQ-VAE
		GAN-based					
		[255]	TGANs-c	2018	Temporal GAN conditioned on the caption (TGAN-c), 3-discriminators (video, frame, motion), training at video-level and frame-level with temporal coherence loss	description video: 48 × 48; frames: 16	BiLSTM-words + LSTM-sentence Deconv-cGAN (3-discriminators: video, frame, motion)
		[256]	IRC-GAN	2019	Recurrent transconvolutional generator (RTG) having LSTM cells with 2D transConv net, Mutual-information introspection (MI) for semantic similarity in 2 stages	description video: 64 × 64; frames: 16	one-hot-vector + BiLSTM + LSTM-encoder LSTM + TransConv2D + cGAN
		[257]	TFGAN	2019	Multi-scale text-conditioning on the discriminative convolutional filter, a new synthetic dataset for text–video modality	description video: 128 × 128; frames: 16	CNN + GRU-recurrent modified-MoCoGAN
		[258]	Latent-Path	2021	Introduced T2V generation on a real dataset, discriminator with single-frame (2D-Conv) and multi-frame (3D-Conv), and Stacked-pooling block for generating frames from latent representations	description video: 64 × 64; frames: 6, 16	BERT 2D/3D-CNN + stacked-upPooling
		[259]	TiVGAN	2021	Text-to-image-to-video GAN (TiVGAN) framework, 2-stage model (T2I and frame-by-frame generation), training stabilization techniques (independent sample pairing, 2-branch discriminator)	description=>image video: 128 × 128; frames: 22	Skip-thought+PCA GAN-INT-CLS+GRU
	Manipulation	GAN-based					
		[260]	M3L	2022	Introduced language-based video editing task (LBVE), Multi-modal multi-level transformer for text–video editing, 3 new datasets (E-MNIST, E-CLEVR, E-JESTER)	description+video video: 128 × 128; frames: 35	RoBERTa [261] 3D ResNet

**Table 3 sensors-22-06816-t003:** Datasets found in the selected paper.

Name	Year	Designed for	Source	Stats						Annotations
				**Approx. Size** (GB)	Quantity				Quality	
					Training	Validation	Testing	Total	Approx. Resolution (px)	
Image data										
Animal datasets										
AwA2 [316]	2018	Transfer-learning	AwA [317], Internet (Flicker, Wikipedia)	13	20,142	9698	7460	37,300	-	85 binary-continuous class attributes
AFHQ [318]	2021	Image-to-Image translation	Flicker, Pixabay	0.3	13,500	-	1500	15,000	512 × 512	3-domain (cat, dog, wildlife), breed information
Digit datasets										
SVHN [319]	2011	Object-recognition, Text–natural_image learning	Google Street View	2.3	604,388	-	26,032	630,420	32 × 32	10 classes, character-level Bounding box, multi-digit representation
MNIST [63]	1998	Pattern-recognition	NIST	0.1	60,000	-	10,000	70,000	28 × 28	0–9 labels, 1 digit/image
MNIST-CB [320]	2018	Pattern-recognition	MNIST	-	50,000	-	10,000	60,000	256 × 256	0–9 labels, 1 digit/image
Color-MNIST [111]	2018		MNIST	-	8000	-	8000	16,000	256 × 256	2 digits/image, 2 sizes, 6 colors, 4 relations
Multi-MNIST [175]	2019		MNIST, AIR [321]	0.202	50,000	-	10,000	60,000	256 × 256	3 digits/image, labels, layout-encoding, split_digits
Object-centric datasets										
Oxford-102 [322]	2009	Image-Classification, Fine-grain Recognition	Internet	0.5	7034	-	1155	8189	-	102 categories, chi2-distance, labels, segmentation-mask, low-level (color, gradient-histogram, SIFT), 10 captions/image [43]
CUB-2010 [323]	2010	Subordinate categorization	Flicker	0.7	3000	-	3033	6033	-	Bounding Box, Rough Segmentation, Attributes, labels, 10 captions/image [43]
black CUB-2011 [323]	2011			1.2	8855	-	2933	11,788	-	200-categories, 15 Part Locations, 312 Binary Attributes, 1 Bounding Box, labels, 10 captions/image [43], text commands [216]
Application datasets										
GRP [324]	2016	Real-world interaction learning	Ten 7-DOF robot arms pushing	137	54,000	1500	1500	57,000	640 × 512, 256 × 256	Robot joint-angle, gripper-pose, commanded gripper-pose, measured torques, images, 3–5 sec videos
Robotic-videos [325]	2018	Visuomotor policies	Camera recordings, commands	4.7	-	-	-	10,003	-	10 fps, avg. 20 sec videos, 3 angles, 3 cameras, attention map, pick–push task
Facial datasets										
LFW [326]	2007	Face-recognition	Faces-in-the-wild [327], Viola-Jones [328]	1.5	2200	-	1000	13,233 ~total	250 × 250	4 categories (original, 3 aligned), labels, names
CelebA [329]	2015	Facial-attribute learning	CelebFaces [330]	23	160,000	20,000	20,000	200,000	Original, 218 × 178	Bounding boxes, Landmarks, Attributes, Identity, text commands [216]
CelebA-HQ [281]	2018	High-quality Facial-learning	CelebA	28	-	-	-	30,000	1024 × 1024	+ high quality
FFHQ [157]	2019	Facial-learing	Flicker, MFA-ERT [331]	1280	60,000	10,000	-	70,000	Original, 1024 × 1024, 128 × 128	unsupervised high-level face attributes
CelebTD-HQ [155]	2020	Text-to-faces	Celeb-HQ	-	24,000	-	6000	30,000	1024 × 1024	+ 10 descriptions/image
Multi-modal CelebA-HQ [229]	2021	Text-guided Multi-modal generation		20	24,000	-	6000	30,000	1024 × 1024, 512 × 512	+ 10 descriptions/image, label map, sketches
Long-text datasets										
VQA [332]	2015	Visual-reasoning	MS-COCO, Abstract-Scenes	-	102,783	50,504	101,434	254,721	-	5 captions, 3 questions/image, 10 answers/image
VQA-2.0 [288]	2017		MS-COCO, Abstract-Scenes, Binary-Abstract-Scenes [333]	-	443,000	214,000	453,000	1,110,000	-	+ 3 question/image, 10 answer/question, image–question–answer pair
Recipe1M [334]	2017	High-capacity Multi-modal learning	~24 cooking websites	135	619,508	133,860	134,338	887,706	-	1M recipes (ingredients + instructions), title, labels
Synthetic datasets										
Abstract Scenes [335]	2013	Semantic-information (vision-language)	58-category clip-arts	0.8	8016	-	2004	10,020	-	58 classes, person attributes, co-occurrence, absolute spatial location, relative spatial location, depth ordering
CoDraw [336]	2019	Goal-driven human–machine interaction	Abstract-Scenes, LViS [94], VisDial [285]	1	7989	1002	1002	9993	-	+ dialogues, utterance–snapshot pairs
CLEVR [337]	2016	Visual-reasoning	Computer-generated CLEVR-universe	19	70,000	15,000	15,000	100,000	224 × 224 ~unclear	Q-A, Scene-graphs, Functional-program
i-CLEVR [169]	2019			10	30,000	10,000	10,000	50,000	-	+ sequence of 5 image–instruction pairs
CLEVR-G [111]	2018		CLEVR-(256 × 256) images	0.06	10,000	-	10,000	20,000	256 × 256	+ still images
CLEVR-SV [231]	2019	Text-to-visual story	CLEVR	-	10,000	-	3000	13,000	320 × 240	4 objects/story, metallic/rubber objects, 8 colors, 2 sizes, 3 shapes
Anime [338]	2021	Machine-learning	DANBOORU-2021 [339]	265	-	-	-	1,213,000	512 (w,h)	hand Bounding boxes, faces, figures, hand
Real-world datasets										
PASCAL-VOC2007 [340]	2007	Object-detection, Classification, Segmentation	Flicker	1	2501	2510	5011	10,022		2 classes, viewpoint, Bounding box, occlusion/truncation, difficult, segmentation (class, object), person layout, user tags [341]
black MIR-Flicker25k [342]	2008	Classification, Retrival		3	15,000	-	10,000	25,000	Original	multi-level labels, manual tags, EXIF
MIR-Flicker-1M [343]	2010			12	-	-	-	100,000	Original, 256 × 256	+ “user-tags”, Pyramid histogram of words [344], GIST [345], MPEG-7 descriptors [346]
black CIFAR-10 [347]	2009	Image generation	Internet (Google, Flicker, Altavista), WordNet [348]	0.2	50,000	-	10,000	60,000	32 × 32	10 classes, labels
LSUN [349]	2015		Google-images, Amazon-Mechanical-Turk (AMT), PASCAL-VOC-2012 [350], SUN [351]	1736	-	-	-	60,000,000	256 (w,h)	10 scenes, 20 objects, labels
YFCC100M [352]	2016	Computer vision	Flicker	15	-	-	-	99,206,564 (image), 793,436 (video)	-	user tags, pictures, and videos, geographic location, extraction timespan, camera info
black ILSVRC: ImageNet [353]	2017	Classification, Retrival, Detection, Feature extraction	Internet, WordNet [348]	166	1,281,167	50,000	100,000	14,197,122 ~total	400 × 350	Bounding boxes, SIFT features, labels, synets
MS-COCO [354]	2015	Detection, segmentation	Flickr	25	165,482	81,208	81,434	328,124	-	Pixel-level segmentation, 91 object classes, 5 descriptions, panoptic, Instance spotting, Bounding boxes, Keypoint detection, dense pose, VisDial dialogue, Scene graphs
COCO-stuff [355]	2018	Background in computer vision	MS-COCO	21	118,490	5400	40,900	164,790	-	+ stuff_labels
LN-COCO [294]	2020	Multimodal tasks (vision–language), image captioning		7	134,272	8573	-	142,845	-	captions, speech, groundings (mouse-trace)
CC3M [356]	2018	Image-captioning	Flumejava [357]	0.6	3,318,333	28,000	22,500	3,368,833	400 (w,h)	image–caption pair, labels,
VG [197]	2017	Cognitive-task	MS-COCO, YFCC100M	15	-	-	-	108,077	500~width	Region-descriptions, Objects, Attributes, Relationships, Region-graphs, Scene-graphs, Q-A pairs
black VG+ [187]	2020		VG	-	-	-	-	217,000	-	-
OpenImages [358]	2020	Image classification	Flicker	565	9,011,219	41,620	125,436	9,178,275	1600 × 1200, 300,000-px	Class-labels, image-labels, Bounding boxes, visual relation annotation
LN-OpenImages [294]	2020	Multimodal tasks (vision-language)	OpenImages	21	507,444	41,691	126,020	675,155	-	captions, speech, groundings (mouse-trace)
LAION-400M [359]	2021	Multi-modal Lanuage-vision learning	Common-Crawl [360]	11,050	-	-	-	413,000,000	1024, 512, 256	image–caption pair
3D datasets										
Primitive Shapes [120]	2018	Text-to-3D_shape	Voxilizing 6-type primitives	0.05	6048	756	756	7560	32 × 32	synthetic 255 descriptions/primitive, 6 shape labels, 14 colors, 9 sizes
ShapeNetCore [120]			ShapeNet [361], AMT	11	12,032	1503	1503	15,038	32 × 32, 256 × 256, 128 × 128	5 descriptions/shape, color voxelization (suface, solid), 2 categories (table, chair)
Text–to–3D House Model [122]	2020	House-planning	-	1	1600~houses, 503~textures	-	400~houses, 370~textures	2000~houses, 873~textures		avg. 6 rooms/house, 1 description, textures_images
IScene [247]	2020	Text-to-3D video generation	Computer generated	-	100,000~static 100,000~animated	5000~static 5000~animated	6400~static 6400~animated	1,300,000~static 1,400,000~animated	-	13 captions/static scene, 14 captions/animated
Editing datasets										
ReferIt [362]	2014	Natural language referring the expression	ImageCLEF IAPR [363], SAIAPR TC-12 [364]	3	10,000	-	9894	19,894	-	238 object categories, avg. of 7 descriptions/image, labels, segmenation maps, object attributes
black Fashion-synthesis [365]	2017	Text-based image editing	DeepFashion	8	70,000		8979	78,979	256 × 256	descriptions, labels (gender, color, sleeve, category attributes), segmentation maps, Bounding boxes, dense pose, landmark
CoSaL [204]	2018	Language-Based Image Editing	Computer-generated	-	50,000	-	10,000	60,000	-	9 shapes, descriptions (direct, relational)
Global-edit-Data [226]	2018	Global Image editing	AMT, MIT-Adobe-5K [366]	-	1378	252	252	1882	-	original edit pair, transformation rating, phrase description of transformation
Zap-Seq [172]	2020	Interactive image editing	UT-Zap50K [367]	-	-	-	-	8734	-	3–5 image sequences, shoes, attributes, multi-captions
DeepFashion-Seq [172]	2020		Deepfashion [368]	-	-	-	-	4820	-	clothes, attributes, 3-5 image sequences, multi-captions
GIER [205]	2020	Language-Based Image Editing	Zhopped.com [369], Reddit.com [370], AMT, Upwork	7.5	4934	618	618	6170	128 × 128, 300 × 500	5 language_requests, 23 editing operations, masks
MA5k-Req [207]	2021	Image editing	AMT, MIT-Adobe-5K [366]	9.5	17,325	2475	4950	24,750	-	5 edits/image, 1 description/image
Story datasets										
VIST [371]	2016	Sequential vision-to-language	YFCC100M, AMT, Stanford CoreNLP [227]	320	40,108	5013	5013	50,136~stories	-	5-image/story, 1-caption/image
PororoQA [372]	2017	Visual question answering	Pororo, AMT	11.5	103~episodes, 5521~QA	34~episodes, 1955~QA	34~episodes, 1437~QA	171~episodes, 8913~QA	-	40-s video/story (408-movies), multi-captions/1-s video, multi-QA/story, 13-characters
black Pororo-SV [231]	2019	Text-to-visual story	PororoQA	-	13,000	-	2336	15,336	-	1-description/story, 5-image/story
Video datasets										
TRECVID’03 news [373]	2003	Video information retrieval	ABC World News Tonight, CNN headline news, C-SPAN programs	-	127~h	-	6~h	133~h	-	Story segmentation, 17 features, shot separations, (1894 binary word/ video-shot, 166 HSV color correlogram [92])
black FlintstonesSV [239]	2021	Sequential vision-to-language	FlintStones Dataset	5	20,132	2071	2309	24,512	-	+ 7 characters, 5 images/story
FlintStones Dataset [243]	2018	Video caption perceptual reasoning, semantic scene generation	Flintstones, AMT	128	20,148	2518	2518	25,184	-	3 sec clip (75 frames), Bounding boxes, segmentation maps, 1-4 sentence descriptions/video, clean background, labels
MSVD [374]	2011	Machine paraphrasing	Youtube, AMT	1.7	1773	-	197	1970	-	avg. 40 descriptions/video, 4–10 sec video, multi-lingual descriptions
MSR-VTT [375]	2016	Text–video embedding	Internet, AMT	6	6513	497	2990	10,000	original, 320 × 240	30 fps, 20 captions/video, 41.2 h video, 20 categories
Text-to-video-dataset [253]	2017		Youtube, KHAV [376]	-	2800	400	800	4000	original, 256 × 256	SIFT-keypoints, 25 fps, 10 categories, 400 videos/category, title and description
Epic-Kitchen [377]	2018	Egocentric Vision	Camera recordings, AMT, Youtube caption tool	740	272	-	106~seen, 54~unseen	432	1920 ×1080	60 fps, object bounding boxes, action segmentation, multi-lingual sound recordings, 1–55 min variable duration
Howto100M [310]	2019	Text–video embedding	Youtube, WikiHow	785	-	-	-	1,220,000	original, 256 (w,h)	caption, avg. 110 clip–caption pairs/video, 12 categories
Moving Shapes (v1,v2) [257]	2019		Computer-generated	-	129,200	-	400	129,600	256 × 256	3 shapes, 5 colors, 2 sizes, 3 motion types, 16 frames/video, 1 caption/video
Bouncing MNIST [250]	2017	Text–video generation	Bouncing MNIST	-	10,000	2000	-	12,000	256 × 256	single-digit, 2-digit, labels, caption, 10 frames/video
Video editing datasets										
E-CLEVR [260]	2022	Text-based video editing	CLEVR, CATER [378]	-	10,133	-	729	10,862	128 × 128	20 fps, avg. 13 words/caption, source target video
E-JESTER [260]	2022		20BN-JESTER [379], AMT	-	14,022	-	885	14,907	100 × 176	4 fps, 27 classes, avg. 10 words/caption
E-MNIST [260]	2022		moving-MNIST	-	11,070	-	738	11,808	256 × 256	Source target video, 2 types (S-MNIST, D-MNIST), 30 fps, avg. 5.5 word/caption,
Human action datasets										
MHP [380]	2014	Pose estimation	YouTube videos	13	28,821	-	11,701	40,522	-	body–joint positions, torso–head 3D orientations, joint and body part occlusion labels, 491 activity labels, 3 captions/image [381]
black KTH-Action [382]	2004	Action recognition	Camera recordings	-	770	766	855	2391	160 × 120	6 actions, 25 people, 25 fps, 4 sec video, 4 scenarios, caption-SyncDRAW [250], caption-KTH-4 [256]
black MUG [383]	2010	Facial understanding	Camera images	38	-	-	-	204,242	896 × 896	86 subjects, 80 facial landmarks, 7 emotions, 19 fps, direct emotions FACS [384], video induced emotions
black UCF-101 [314]	2012	Human action recognition	UCF50 [385], YouTube	127	13,320	2104	5613	21,037	320 × 240	101 classes in 5 types, STIP features, 7.2 sec video avg., 25 fps, 25 groups/action, dynamic background, Bounding boxes, class attributes
black A2D [386]	2015		Youtube	20	3036	-	746	3782	-	avg. 136 frames, 7 actors, 8 actions, instance-level segmentation, descriptions [313], frame-level BBox [387]
black KHAV [376]	2017		Youtube, AMT	-	253,540	17,804	34,901	306,245	variable	400 classes, min 400 videos/class, avg. 10 sec video
CUHK-PEDES [388]	2017	Person searching (video surveillance)	CUHK03 [389], Market-1501 [390], AMT, SSM [391], VIPER [392], CUHK01 [393],	-	34,054	3078	3074	40,206	-	2- descriptions/image, attribute labels, orientation phrase [219]

**Table 4 sensors-22-06816-t004:** Comparison of different image models based on the most-used evaluation metrics. Models marked in dark blue represent additional metrics not listed in the table, whereas light blue shows the models which do not give any of these metrics in their own paper. To efficiently compact the table, the symbol “;” separates the datasets within the same row, “-” for no data available, and “,” for listing.

	Evaluation Metrics							
	Quality				Semantics			
Model (Categories of image models)	**Datasets:** (1) Oxford (2) CUB (3) MS-COCO; COCO-stuff (4) CoDraw/Abstract-Scenes (5) Conceptual captions		(6) FFHQ; CelebTD-HQ; CelebA-HQ; CelebA; MM-Celeb-HQ (7) ImageNet (8) CIFAR-10 (9) Visual-genome (10) Fashion-data (Zap-seq; DeepFashion-seq; Fashion-synthesis)		(11) Pororo (12) 3D-houses (13) LN-data (COCO; OpenImages) (14) VQA 2.0 (15) CLEVR		(16) Editing-data (GIER; MA5k-Req; MIT-Adobe5k) (17) CUHK-PEDES (18) Video-generation (KTH; MSVD; MSR-VTT; Kinetics, MUG; UCF-101; A2D)	
	IS (higher-better)	FID, SceneFID (low-better)	SSIM (higher-better)	LPIPS (High-better)	SOA-c, SOA-i % (High-better)	VS-Similarity (High-better)	R-precision % (High-better)	Captioning metrics (BLEU, METEOR, ROUGE_L, CIDEr, SPICE, CapLoss) (High-better)
[100]	(3) 24.77 ± 1.59							(3) (0.614, 0.426, 0.300, 0.218), 0.201, 0.457, 0.656, 0.130
[101]	(2) 1.35 ± 0.25 (3) 17.9 ± 0.15	(2) 56.10 (3) 27.50						
[103]	(3) 18.2	(3) 23.6						(3) -, -, -, -, -, 2.43
[105]	(5) 15.27± 0.59 (6) 4.49 ± 0.05	(5) 22.61 (6) 10.81 (7) 21.19					(6) (CLIP: 0.23 ± 0.03)	
[106]	(1) 4.28 ± 0.09 (2) 6.89 ± 0.06		(1) 0.2174 (2) 0.3160					
[107]	(1) 4.66 ± 0.07 (2) 7.94 ± 0.12		(1) 0.2186 (2) 0.3176					
[5]			(3) 0.156 ± 0.11					
[110]	(1) 4.21 ± 0.06 (2) 4.97 ± 0.03							
[113]		(1) 14.1 (2) 10.32 (3) 13.86 (6) 6.33 (7) 11.89						
[116]	(8) 8.09 ± 0.07							
[7]	(1) 4.17 ± 0.07 (2) 5.08 ± 0.08 (3) 7.88 ± 0.07	(1) 79.55 (2) 68.79 (3) 60.62	(1) 0.1948 (2) 0.2934 (15) 0.596			(2) 0.082 ± 0.147		(3) 0.077, 0.122, -, 0.160
[117]	(8) 8.25 ± 0.07	(12) 220.18						
[118]	3.45 ± 0.05							
[122]		(12) 145.16						
[124]	(2) 5.10	(2) 14.81 (3) 21.42 (6) -; -; -; -; 137.60		(6) -; -; -; -; 0.581			(3) (CLIP: 66.42 ± 1.49)	(3) -, -, -, -, -, 3.09
[32]	(1) 3.20 ± 0.01 (2) 3.70 ± 0.04 (3) 8.45 ± 0.03; 8.4 ± 0.2 (7) 8.84 ± 0.08 (9) 7.39 ± 0.38 (10) 7.88; 6.24	1) 55.28 (2) 51.89 (3) 74.05; 78.19 (7) 89.21 (9) 77.95 (10) 60.62; 65.62	(1) 0.1837 (2) 0.2812 (10) 0.437; 0.316			(1) 0.278 ± 0.134 (2) 0.228 ± 0.162	(2) 10.37 ± 5.88	(3) 0.089, 0.128, -, 0.195; 0.062, 0.095, -, 0.078
[33]	(1) 3.26 ± 0.01 (2) 4.04 ± 0.05 (3) 8.30 ± 0.10 (6) -; -; -; 1.444 (7) 9.55 ± 0.11	2) 15.30 (3) 81.59 (6) -; -; -; 285.48 (7) 44.54 (12) 188.15		(2) 0.028 ± 0.009 (6) -; -; -; 0.292 ± 0.053			(2) 45.28 ± 3.72 (3) 72.83 ± 3.17	
[125]	(2) 3.00 ± 0.03							
[126]	(1) 3.45 ± 0.07 (2) 4.15 ± 0.05 (3) 11.86 ± 0.18; 11.9 ± 0.2	(1) 40.02 ± 0.55 (2) 18.23 (3) 75.34	(1) 0.1886 (2) 0.2887			(1) 0.296 ± 0.131 (2) 0.246 ± 0.157 (3) 0.199 ± 0.183		
[127]	(1) 3.52 ± 0.02 (2) 4.38 ± 0.05					(1) 0.297 ± 0.136 (2) 0.290 ± 0.149		
[128]	(1) 3.57 ± 0.05 (2) 4.48 ± 0.04 (3) 27.53 ± 0.25					(1) 0.303 ± 0.137 (2) 0.253 ± 0.165 (3) 0.227 ± 0.145		
[34]	(1) 3.55 ± 0.06 (2) 4.36 ± 0.03 (3) 25.89 ± 0.47; 25.9 ± 0.5 (9) 8.20 ± 0.35 (10) 9.79; 8.28 (13) 20.80; 15.3 (14) 20.53 ± 0.36 (17) 3.726 ± 0.123	(2) 23.98 (3) 35.49; 35.49 (6) -; -; -; -; 125.98 (9) 72.11 (10) 48.58; 55.76 (13) 51.80; 56.6 (14) 44.35	(1) 0.1873 (2) 0.3129 (10) 0.527; 0.405 (17) 0.298 ± 0.126	(6) -; -; -; -; 0.512	(3) 25.88, 38.79	(2) 0.279 (3) 0.071	(1) 20.3 ± 1.5 (2) 67.82 ± 4.43 (3) 85.47 ± 3.69 (CLIP: 65.66 ± 2.83) (13) 43.88	(3) -, -, -, 0.695 ± 0.005, -, 3.01; 0.087, 0.105, -, 0.251
[129]	(3) 23.74 ± 0.36	(3) 34.52					(3) 86.44 ± 3.38	
[130]	(2) 4.67 ± 0.04 (3) 27.86 ± 0.31	(2) 18.167 (3) 32.276				(2) 0.302 (3) 0.089		
[131]	(2) 4.58 ± 0.09 (3) 24.06 ± 0.60	(6) -; -; -; -; 116.32		(6) -; -; -; -; 0.522	(3) 25.64, -		(2) 69.33 ± 3.23 (3) 82.43 ± 2.43	
[132]	(2) 4.56 ± 0.05 (3) 25.98 ± 0.04	(2) 10.41 (3) 29.29						
[133]	(1) 3.98 ± 0.05 (2) 4.48 ± 0.05							
[134]	(2) 5.03 ± 0.03 (3) 31.01 ± 0.34	(2) 11.83 (3) 31.97						
[135]	(2) 4.91 ± 0.03 (3) 30.85 ± 0.7	(2) 14.3 (3) 31.14			(3) 32.78, -		(2) 71.57 ± 1.2 (3) 89.57 ± 0.9	(3) 0.381, -, -, -, -
[136]	(2) 4.42 ± 0.04 (3) 35.08 ± 1.16	(2) 15.19 (3) 28.12					(2) 85.45 ± 0.57 (3) 92.61 ± 0.50	
[137]	(1) 3.65 ± 0.06							
[138]	(2) 4.67 ± 0.09 (3) 35.69 ± 0.50; 35.7 ± 0.5	(3) 29.35					(3) 51.68	
[139]	(2) 4.85 ± 0.04 (3) 31.67 ± 0.36	(2) 17.32 (3) 30.73						
[140]	(1) 3.87 ± 0.05 (2) 4.66 ± 0.04	(1) 32.64 (2) 9.35						
[141]	(7) 60.6 ± 1.6							
[143]	(2) 4.56 ± 0.05 (3) 26.47 ± 0.41; 26.5 ± 0.4	(2) 18.34 (3) 34.71			(3) 27.52, -		(2) 60.42 ± 4.39 (3) 80.21 ± 0.39	
[145]	(1) 2.90 ± 0.03 (2) 3.58 ± 0.05 (3) 8.94 ± 0.20	(1) 37.94 ± 0.39 (2) 18.41 ± 1.07 (3) 27.07 ± 2.55						
[146]	(2) 4.75 ± 0.07 (3) 30.49 ± 0.57; 30.5 ± 0.6	(2) 16.09 (3) 32.64; 32.64 (6) -; -; -; -; 131.05		(6) -; -; -; -; 0.544	(3) 33.44, 48.03		(1) 19.9 ± 1.4 (2) 72.31 ± 0.91 (3) 88.56 ± 0.28 (CLIP: 65.45 ± 2.18)	(3) -, -, -, 0.823 ± 0.002, -, 2.87
[148]	(3) 52.73 ± 0.61	(3) 55.82			(3) 77.02, 84.55		(3) 93.59	
[150] (AttnGAN, DM-GAN)	(2) 4.42 ± 0.05, 4.77 ± 0.05 (3) 25.70 ± 0.62, 33.34 ± 0.51	(2) 16.34, 14.38 (3) 23.93, 20.79					(2) 69.64 ± 0.63, 78.99 ± 0.66 (3) 86.55 ± 0.51, 93.40 ± 0.39	
[151]	(3) 30.45 (13) 28.37; 24.90	(3) 9.33 (13) 14.12; 26.91			(3) 50.94, 71.33 (13) 36.76, 48.14		(3) 71.00 (13) 66.92; 57.55	
[152]	(2) 4.74 ± 0.04 (3) 16.40 ± 0.30					(2) 0.298 ± 0.146		
[153]	(1) 3.71 ± 0.06 (2) 4.23 ± 0.05	(1) 16.47 (2) 11.17						
[155]	(2) 4.78 ± 0.03 (3) 33.0 ± 0.31	(6) -; 5.08 ± 0.07					(2) 79.56 (3) 88.23	
[158]	(3) -; 34.7 ± 0.3	(3) -; 30.63						
[159]	(3) 32.88 ± 0.93	(3) 25.24					(3) 63.80 ± 1.12 (CLIP: 98.44 ± 0.15)	
[160]	(2) 5.97 (3) 32.34 (6) -; -; 2.93 (13) 26.32	(2) 10.48 (3) 8.12 (6) -; -; 12.54 (13) 11.78			(3) 61.09, 74.78			
[162]	(1) 3.52 ± 0.15 (2) 4.07 ± 0.13							
[165]	(1) 4.53 ± 0.05 (2) 5.23 ± 0.09						(1) 26.7 ± 1.6 (2) 23.8 ± 1.5	
[167]	(1) 4.09 ± 0.08 (2) 4.76 ± 0.05	(1) 41.85 (2) 21.66						
[168]	(3) 9.74 ± 0.02							
[169]		(16) 87.0128; 33.7366	(16) 0.7492; 0.7772					
[171]	(14) 21.92 ± 0.25	(14) 41.7 (15) 36.14						
[172]	(10) 9.58; 8.41	(10) 50.31; 53.18	(10) 0.651; 0.498					
[173]	(3) 26.64	(3) 25.38					(3) 84.79	
[43]	(2) 3.62 ± 0.07	(2) 67.22	(2) 0.237			(2) 0.114 ± 0.151		
[175] (StackGAN, AttnGAN)	(3) 12.12 ± 0.31, 24.76 ± 0.43	(3) 55.30 ± 1.78, 33.35 ± 1.15						
[177]	(3) 27.88 ± 0.12	(3) 24.70 ± 0.09			(3) 35.85, 50.47		(3) 89.01 ± 0.26	(3) -, -, -, 0.819 ± 0.004
[178]	(3) -; 17.0 ± 0.1 (9) 14.4 ± 0.6	(3) -; 45.96, 16.76 (9) 39.07, 9.63						
[180]	(3) 11.46 ± 0.09; 11.46 ± 0.09							(3) -; 0.122, 0.154, -, 0.367
[184]	(3) 32.79 ± 0.21 (13) 16.5	(3) 21.21 (13) 66.5			(3) 27.14, 41.24		(3) 93.39 ± 2.08	(3) -, -, -, 0.783 ± 0.002
[186]	(1) 3.92 ± 0.02 (2) 4.62 ± 0.06						(1) 85.81 (2) 85.28	
[187]		(3) -; 32.31 (9) 20.83						
[189]	(1) 4.72 ± 0.1 (2) 4.97 ± 0.21 (3) 29.87 ± 0.09						(1) 74.32 (2) 63.78 (3) 79.57	
[191]	(2) 5.06 ± 0.21 (3) 29.03 ± 0.15	(2) 16.87 (3) 20.06					(2) 99.8 (3) 95.0	
[192]	(3) -; 7.3 ± 0.1 (9) 6.3 ± 0.2	(3) -; 67.96 (9) 74.61		(3) -; 0.29 ± 0.10 (9) 0.31 ± 0.08				(3) 0.107, 0.141, -, 0.238
[194]	(3) -; 4.14							
[198]	(3) -; 14.5 ± 0.7	(3) -; 81.0		(3) -; 0.67 ± 0.05				
[199]	(3) -; 10.2 ± 0.2 (9) 8.2 ± 0.2	(3) -; 38.29 (9) 35.25		(3) -; 0.32 ± 0.09 (9) 0.29 ± 0.08				
[200]	(3) -; 14.78 ± 0.65 (9) 12.03 ± 0.37	(3) -; 26.32 (9) 27.33		(3) -; 0.52 ± 0.09 (9) 0.56 ± 0.06				(3) 0.139, 0.157, -, 0.325
[201] (StackGAN, AttnGAN, DM-GAN)	(3) 10.38 ± 0.2, 28.18 ± 0.51, 32.37 ± 0.31	(3) -, 29.26, 32.37					(3) -, 86.39 ± 0.0039, 90.37 ± 0.0063	
[203]	(13) 21.30; 14.7	(13) 48.70; 61.9					(13) 37.88	
[207]		(16) 49.2049; 6.7571	(16) 0.8160; 0.8459					
[208]	(1) 5.03 ± 0.62 (2) 1.92 ± 0.05 (10) -; -; 8.65 ± 1.33 (17) 3.790 ± 0.182	(10) 22.86 (16) 140.1495; 30.9877	(16) 0.7300; 0.7938 (17) 0.239 ± 0.106				(2) 0.045 (3) 0.077	
[210]	(2) 4.451 (6) -; -; -; 1.178 (10) 9.83; 8.26	(2) 50.51 (6) -; -; -; 421.84 (10) 47.25; 56.49 (16) 112.4168; 43.9463	(10) 0.512; 0.428 (16) 0.5777; 0.5429	(2) 0.060 ± 0.024 (6) -; -; -; 0.024 ± 0.012			(2) 0.048 (3) 0.089	
[211]	(1) 4.83 ± 0.48 (2) 2.59 ± 0.11 (10) -; -; 8.78 ± 1.43	(10) -; -; 10.72						
[213]								
[214]	(1) 6.26 ± 0.44 (2) 2.76 ± 0.08 (10) -; -; 11.63 ± 2.15	(16) 214.7331; 102.1330	(16) 0.4395; 0.4988					
[215]	(2) 8.47 (3) 14.96	(2) 11.74 (3) 25.08 (6) 143.39; -; -; -; 117.89		(2) 0.001 ± 0.000			(2) 10.1 (3) 8.7	
[216]	(2) 4.599 (6) -; -; -; 3.069	(2) 2.96 (6) -; -; -; 32.14		(2) 0.081 ± 0.001 (6) -; -; -; 0.152 ± 0.003				
[219]	(17) 4.218 ± 0.195		(17) 0.364 ± 0.123					
[221]		(2) 8.02 (3) 12.39						
[222]			(2) 0.0547 (3) 0.0709					
[226]		(16) 74.7761; 14.5538	(16) 0.7293; 0.7938					
[229] (generation, manipulation)		(6) (-, 135.47); -; -; -; (106.37, 107.25)		(6) -; -; -; -; 0.456				

**Table 5 sensors-22-06816-t005:** Similar to image model evaluation metric table, but represents story and video tasks.

Model	**Evaluation Metrics**														
	**Quality**														
	**Datasets:** (1) Pororo (2) CLEVR (3) Flintstones (4) KTH (5) MSVD (6) MSR-VTT (7) Kinetics (8) MUG							(9) UCF-101 (10) A2D (11) IScene (12) MNIST (1-digit; 2-digit) (13) Text-Video-data (14) Robotic-dataset (15) Video-manipulation (E-MNIST; E-CLEVR; E-JESTER)							
	IS (Frame, Video) (higher-better)	FID (low-better)	FSD (low-better)	FVD (low-better)	SSIM (higher-better)	VAD (Low-better)	GAM (low-diverse)	Accuracy (Object, Gesture, Frame) (High-better)	Char. F1 (High-better)	CA (High-better)	NLL (Low-better)	R-precision % (High-better)	RM (High-better)	CLIP-sim (High-better)	BLEU (High-better)
[230]		(1) 50.24	(1) 30.40					(1) -, -, 28.06	(1) 58.11	58.11					(1) 5.30/2.34
[231]		(1) 49.27	(91) 111.09	(1) 274.59	(1) 0.481 (2) 0.672				(1) 27.0			(1) 1.51 ± 0.15			(1) 3.24/1.22
[31]					(1) 0.509				(1) 25.7						
[233]					(1) 0.521				(1) 38.0						
[235]		(1) 40.56	(1) 71.51	(1) 190.59								(1) 1.76 ± 0.04			(1) 3.25/1.22
[237]		(1) 34.53	(1) 171.36					(1) -, -, 13.97	(1) 38.01			(1) 3.56 ± 0.04			(1) 3.68/1.34
[239]		(1) 18.09						(1) -, -, 17.36	(1) 43.02			(1) 3.28 ± 0.00			(1) 3.80/1.44
[243]											(3) 7.636				
[244]	(4) 2.077 ± 0.299, 1.280 ± 0.024 (5) 2.580 ± 0.125, 1.141 ± 0.013														
[247] (static, animated)					(11) 0.812, 0.849										
[250]											(12) 340.39; 639.71				
[252]										(4) 70.95					
[253]	(7) 82.13, 14.65									(7) 42.6 (13) 42.6					
[254]													(6) 98.34	(6) 24.02	
[255]	(4) 1.937 ± 0.134, 1.005 ± 0.002 (5) 1.749 ± 0.031, 1.003 ± 0.001 (7) -, 4.87 (8) -, 4.65 (9) -, 3.95 ± 0.19 (10) -, 3.84 ± 0.12 (14) -, 2.97 ± 0.21	(4) 69.92 (9) 51.64 (10) 31.56 (14) 6.59					(5) 0.96			(14) 70.4		(9) 0.19 (10) 0.31			
[256]							(4) 0.667 (12) 0.673; 0.687								
[257]	(7) 31.76, 7.19									(7) 76.2					
[258]	(9) -, 7.01 ± 0.36 (10) -, 4.85 ± 0.16 (14) -, 3.36 ± 0.15	(9) 51.64 (10) 25.91 (14) 3.79								(14) 76.6		(9) 0.43 (10) 0.39			
[259]	(7) -, 5.55 (8) -, 5.34	(4) 47.34								(7) 77.8					
[260]						(15) 1.90; 1.96; 1.44		(15) 93.2; 84.5; -, 89.3							

## Data Availability

Not applicable.

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
