# Peer review of "A Review of Multi-Modal Learning from the Text-Guided Visual Processing Viewpoint"

_sensors, 2022, doi:10.3390/s22186816_

Round 1

Reviewer 1 Report

It is always a hard comment, but the reading of the paper has been extremely difficult due to poor English. This meant many of the key contributions of the paper may have not been highlighted well. But a lot of the work was really not understandable.

I recommend a strong edit of the work to make the paper readable. 

The motivation behind the work is definitely valid, but the poor explanation makes it really difficult to judge the work on scientific quality and the insights derived.

Reviewer 2 Report

This paper presents a thorough review of text-to-image synthesis techniques from a broader view, namely text-guided visual-output. Based on this taxonomy, recent SOTA models are analyzed based on a range of datasets. My minor concerns are:
1. The overall presentation, including language, figures, and tables, should be improved. Countless issues can be found, e.g., (1) The title should be capitalized in a consistent way. (2) line 30: an empty pair of brackets following ‘… AI techniques’ (3) Bottom line of Table. 1 is missing. (4) typo in the caption of Figure 6: ‘Deep beleif Network (DBN)’ (5) line 1394: a verb is missing It is highly recommended that this manuscript should be revised carefully by native English speakers, and all the minor issues (not limited to the examples above) should be fixed.

2. In section 2.1.1, reinforcement learning is mentioned on line 110. However, it is not discussed in the following paragraph.

3. Text font in tables, e.g., Table 2 and Table 3, are too small, and the marks (e.g., blue and red highlights in Table 2) are too vague. It is highly recommended that these tables should be reorganized so that they can be fit in a paper.

4. Figure 3 is not mentioned in this manuscript. Moreover, the caption should be something like ‘scheme of GAN’ instead of ‘Vision models’, and the output of ‘Discriminative’ should be a decision (true of false) instead of a regression result.

Reviewer 3 Report

This paper provided a survey on Multi-modal learning from Text-guided Visual processing viewpoint. My major concerns:

1) Figure 2 seem useless, as we all know those common terms.

2) some figures are not informative, such as fig 3, what is input, output? you should give more information on the figure context, rather than listing an abstract figure for readers.

3) I think Fig.4 is not complete, as discriminative models should contain traditonal methods, apart from deep learning methods. So the survey is incomplete.

4) is it possible to further summarize some open-source tools for this task?
